# Phagocytes produce prostaglandin E₂ in response to cytosolic *Listeria monocytogenes*

**Courtney E. McDougal**[1,2], **Zachary T. Morrow**[1], **Tighe Christopher**[1], **Seonyoung Kim**[3], **Drake Carter**[3], **David M. Stevenson**[4], **Daniel Amador-Noguez**[4], **Mark J. Miller**[3], **John-Demian Sauer**[1]*

**1** Department of Medical Microbiology and Immunology, University of Wisconsin-Madison, Madison, Wisconsin, United States of America, **2** Microbiology Doctoral Training Program, University of Wisconsin-Madison, Wisconsin, United States of America, **3** Department of Internal Medicine, Division of Infectious Diseases, Washington University School of Medicine, St. Louis, Missouri, United States of America, **4** Department of Bacteriology, University of Wisconsin-Madison, Madison, Wisconsin, United States of America

* sauer3@wisc.edu

**Data Availability Statement:** All relevant data are within the manuscript and its Supporting Information files.

## Abstract

*Listeria monocytogenes* is an intracellular bacterium that elicits robust CD8+ T-cell responses. Despite the ongoing development of *L. monocytogenes*-based platforms as cancer vaccines, our understanding of how *L. monocytogenes* drives robust CD8+ T-cell responses remains incomplete. One overarching hypothesis is that activation of cytosolic innate pathways is critical for immunity, as strains of *L. monocytogenes* that are unable to access the cytosol fail to elicit robust CD8+ T-cell responses and in fact inhibit optimal T-cell priming. Counterintuitively, however, activation of known cytosolic pathways, such as the inflammasome and type I IFN, lead to impaired immunity. Conversely, production of prostaglandin E₂ (PGE₂) downstream of cyclooxygenase-2 (COX-2) is essential for optimal *L. monocytogenes* T-cell priming. Here, we demonstrate that vacuole-constrained *L. monocytogenes* elicit reduced PGE₂ production compared to wild-type strains in macrophages and dendritic cells *ex vivo*. *In vivo*, infection with wild-type *L. monocytogenes* leads to 10-fold increases in PGE₂ production early during infection whereas vacuole-constrained strains fail to induce PGE₂ over mock-immunized controls. Mice deficient in COX-2 specifically in Lyz2+ or CD11c+ cells produce less PGE₂, suggesting these cell subsets contribute to PGE₂ levels *in vivo*, while depletion of phagocytes with clodronate abolishes PGE₂ production completely. Taken together, this work demonstrates that optimal PGE₂ production by phagocytes depends on *L. monocytogenes* access to the cytosol, suggesting that one reason cytosolic access is required to prime CD8+ T-cell responses may be to facilitate production of PGE₂.

## Author summary

*L. monocytogenes* is an intracellular bacterial pathogen that generates robust cell-mediated immune responses. Due to this robust induction, *L. monocytogenes* is used as both a

**Funding:** The study was supported by the National Institutes of Health (R01CA188034 to JDS, R01AI077600 to MJM, and T32AI55397 to CEM) - www.nih.gov. Additionally this study was supported by the National Science Foundation Graduate Research Fellowship Program (DGE-1747503 to ZTM) - https://www.nsfgrfp.org/. The funders had no role in study design, data collection and analysis, decision to publish, or preparation of the manuscript.

**Competing interests:** The authors have declared that no competing interests exist.

model to understand how CD8$^+$ T-cells are primed, as well as a platform for cancer immunotherapy vaccines. *L. monocytogenes* must enter the cytosol of an infected host cell to stimulate robust T-cell responses, however, which cytosolic innate pathway (s) contribute to T-cell priming remains unclear. Previous data demonstrated that COX-2-dependent PGE$_2$ production is critical for T-cell responses to *L. monocytogenes*. Here, we found that *ex vivo* PGE$_2$ production by macrophages and dendritic cells is partially dependent on cytosolic access, as vacuole-constrained strains of *L. monocytogenes* elicit reduced PGE$_2$. *In vivo*, cytosolic access is essential for PGE$_2$ production. *L. monocytogenes* elicits a 10-fold increase in PGE$_2$ production, whereas strains of *L. monocytogenes* that cannot access the cytosol fail to elicit PGE$_2$ compared to mock immunized mice. Furthermore, CD11c$^+$ and Lyz2$^+$ cells contribute to PGE$_2$ production *in vivo*, as mice deficient in COX-2 in these cell subsets have impaired PGE$_2$ production, while mice depleted of all phagocytes by clodronate treatment are incapable of producing PGE$_2$. Taken together, our work furthers our understanding of how PGE$_2$, a molecule critical for generating T-cell responses, is generated during immunization with *L. monocytogenes*.

## Introduction

*Listeria monocytogenes* is a Gram-positive, intracellular pathogen that elicits robust T-cell responses. Though both CD4$^+$ and CD8$^+$ T-cells contribute to development of *Listeria*-induced immunity, experiments involving adoptive transfers or *in vivo* depletion highlight the importance of CD8$^+$ T-cells specifically in mediating protective responses [1,2]. As such, *L. monocytogenes* has been used for decades as a model to understand how CD8$^+$ T-cell responses are primed [3]. Understanding these responses has become more pressing recently as *Listeria*-based platforms aiming to drive CD8$^+$ T-cell responses are in clinical trials as cancer immunotherapies [4]. Initial work showed that critical signals promoting *Listeria*-stimulated T-cell responses are provided acutely, as bacterial clearance with antibiotics as early as 24 hours-post infection has minimal impact on the kinetics of CD8$^+$ T-cell responses [5]. This work highlights the role of early signals in informing *Listeria*-stimulated cell mediated adaptive responses.

One early signal impacting T-cell responses is the inflammatory environment induced during infection. The importance of the inflammatory milieu on priming T-cell responses has been solidified by multiple groups using antigen-pulsed, matured dendritic cells in combination with non-antigen expressing *L. monocytogenes* as an inflammatory boost [6,7]. These studies enable discrimination between antigen presentation and inflammation and demonstrate that wild-type *L. monocytogenes* provides an optimal inflammatory milieu to drive T-cell priming [6,7]. The inflammatory boost provided through wild-type *L. monocytogenes* infection led to increased T-cell responses, whereas use of strains that specifically alter the inflammatory milieu led to suboptimal responses [6,7]. *L. monocytogenes* activates a number of innate pathways that contribute to the inflammatory milieu. In particular, multiple groups have focused on the role of various cytosolic innate immune pathways, as previous research demonstrated the necessity of cytosolic access in priming cell-mediated immunity [8–10]. *L. monocytogenes* utilizes a cytolysin, listeriolysin O (LLO), to escape from phagosomes directly into the cytosol and LLO-deficient mutants that are unable to access the cytosol inhibit T-cell priming and generate tolerizing immune responses [11,12]. Despite the importance of cytosolic access for priming T-cell responses, multiple cytosol-dependent innate pathways are counterintuitively detrimental to immunity, including STING-dependent type I interferon [13,14] as well as

inflammasome activation [6,15]. We recently identified an alternative innate pathway, production of the eicosanoid prostaglandin E$_2$ (PGE$_2$), as important for immunity as mice deficient in PGE$_2$ have impaired acute and protective T-cell responses to *L. monocytogenes* [16]. Whether PGE$_2$ production is dependent on cytosolic access of *L. monocytogenes* remains unknown, as is which cells produce PGE$_2$ in response to *L. monocytogenes* infection.

Eicosanoids are lipid mediators of inflammation that have potent and diverse biological functions. A major subset of these lipids, including PGE$_2$, are derived from arachidonic acid [17]. During inflammation, arachidonic acid is liberated from the membrane by the cytosolic phospholipase A2 (cPLA2) and then is further metabolized by a number of enzymes including the P450 epoxygenase, lipoxygenases, and cyclooxygenases (COXs) [17]. During infection, PGE$_2$ is produced downstream of COX enzymes, particularly downstream of cyclooxygenase-2 (COX-2) [18]. COX-2 is induced during inflammation and functions to reduce arachidonic acid to prostaglandin H$_2$ (PGH$_2$) [19,20]. PGH$_2$ is further metabolized into different prostaglandins by terminal prostanoid synthases. Coupling of COX enzymes with prostanoid synthases ultimately dictates which prostaglandin will be produced [19,20]. PGE$_2$ specifically is produced by three different terminal synthases, the cytosolic prostaglandin E synthase (cPGES) and microsomal prostaglandin E synthases-1 and -2 (mPGES-1 and mPGES-2) [18,21]. Of these synthases, mPGES-1 is inducible and associated with infection due to its role in inflammatory responses [18,21]. For example, mice deficient in mPGES-1 have reduced febrile and pain responses [18,21]. Previously, we showed that *L. monocytogenes* infection of mice deficient in mPGES-1 or use of a COX-2-specific inhibitor leads to impaired T-cell responses that could be rescued by exogenous dosing of PGE$_2$[16]. Together, these data suggest that production of PGE$_2$ downstream of COX-2 and mPGES-1 is critical for immunity.

During *L. monocytogenes* infection, the cell types responsible for producing PGE$_2$ remain unclear. *L. monocytogenes* is initially captured by a wide range of phagocytic antigen presenting cells (APCs) in the marginal zone of the spleen [22]. Initially, *L. monocytogenes* highly infects multiple macrophage subsets, including MOMA$^+$ metallophilic and MARCO$^+$ marginal zone macrophages [22]. Later, *L. monocytogenes* infection transitions to splenic CD11c$^+$ and CD11b$^+$ cells in the white pulp [22]. Importantly, PGE$_2$ is produced at high amounts early in the immune response, starting at four hours post-immunization and peaking at twelve hours, early timepoints during which macrophages and dendritic cells are heavily infected [16]. Furthermore, one previous study demonstrated that peritoneal macrophages are capable of producing PGE$_2$ after *ex vivo* infection with *L. monocytogenes*[23]. Further analysis is required to elucidate whether macrophages and dendritic cells similarly produce PGE$_2$ *in vivo*.

Here, we demonstrated *ex vivo* that macrophages and dendritic cells produce PGE$_2$ in response to *L. monocytogenes* infection. Importantly, induction of PGE$_2$ *ex vivo* was partially dependent on cytosolic access, as infection of bone marrow-derived macrophages or dendritic cells with vacuole-constrained *L. monocytogenes* led to reduced PGE$_2$ compared to wild-type strains. In contrast, *in vivo* PGE$_2$ production requires cytosolic access, as infection with LLO-deficient *L. monocytogenes* led to a complete lack of PGE$_2$ induction, similar to mock-immunized levels. Lyz2$^+$ and CD11c$^+$ cells contribute to PGE$_2$ production *in vivo*, as deletion of COX-2 selectively in these subsets led to reduced splenic PGE$_2$ levels. However, these subsets are not solely responsible for production as a small amount of PGE$_2$ remains and this remaining PGE$_2$ is sufficient to facilitate optimal T-cell priming. Use of phagocyte-depleting clodronate treatment completely eliminated PGE$_2$ production to mock-immunized levels. Taken together, we show that phagocytes, particularly macrophages and dendritic cells, produce PGE$_2$ in a cytosol-dependent manner.

## Results

### Unprimed macrophages and dendritic cells upregulate PGE$_2$-synthesizing enzymes in response to cytosolic *L. monocytogenes*

We previously demonstrated that immunization of mice with *L. monocytogenes* induces production of PGE$_2$, starting at four hours post-infection and peaking at twelve hours, and that this transient PGE$_2$ production is necessary for optimal T-cell priming [16]. During infection, *L. monocytogenes* infects multiple phagocytic cell populations in the spleen, the majority of which are macrophage and dendritic cell subsets [22]. Initially, *L. monocytogenes* localizes to multiple macrophage subsets [22] and by twelve hours after infection, CD11c$^+$ dendritic cells comprise the largest subset of *L. monocytogenes* infected cells [22]. We hypothesized that macrophages and dendritic cells were the subsets producing PGE$_2$ due these cells being the predominantly infected cell subsets at these early timepoints post immunization. To determine if *L. monocytogenes* infection induces the genes necessary for PGE$_2$ production, we first measured expression of *Pla2g4a* mRNA (encoding cPLA2), a phospholipase that releases arachidonic acid from the cell membrane [17], in bone marrow derived macrophages (BMDMs) and bone marrow derived dendritic cells (BMDCs). BMDMs and BMDCs were infected with *L. monocytogenes* and mRNA was harvested six hours later. We found that *Pla2g4a* expression did not change during *L. monocytogenes* infection (S1A Fig). This result was not surprising, as much of cPLA2 activity is modulated by calcium influx and MAPK phosphorylation rather than transcriptional changes [24]. We next measured mRNA expression of *Ptgs2* (encoding COX-2) and *Ptges* (encoding mPGES-1), enzymes involved in the next steps of PGE$_2$ synthesis [18,21]. In both BMDMs and BMDCs, infection with wild-type *L. monocytogenes* led to an increase in *Ptgs2* expression and, to a lower extent, *Ptges*, suggesting that macrophages and dendritic cells could be capable of synthesizing PGE$_2$ (Fig 1A and 1B). Given that PGE$_2$ is necessary for optimal T-cell priming and that immunizing mice with a strain of *L. monocytogenes* that cannot access the cytosol leads to abolished T-cell effector function [11,12], we hypothesized that the lack of T-cell responses to vacuole-constrained bacteria may be partially due to reduced expression of PGE$_2$-synthesizing enzymes and ultimately decreased production of PGE$_2$. To test this hypothesis, we infected BMDMs and BMDCs with a vacuole-constrained strain of *L. monocytogenes* (Δ*hly*, a mutant lacking the pore-forming protein LLO) and assessed expression of *Ptges* and *Ptgs2* mRNA. Consistent with this hypothesis, infection with this strain led to reduced *Ptgs2* expression in BMDMs and BMDCs, suggesting that cytosolic access is required for optimal expression of *Ptgs2* (Fig 1A). Interestingly, infection with Δ*hly L. monocytogenes* led to similar levels of *Ptges* expression in both BMDMs and BMDCs (Fig 1B). Taken together, these results suggest that cytosolic access increases *Ptgs2* expression in BMDMs and BMDCs, whereas *Ptges* expression is induced independently of cytosolic access. Additionally, as controls we assessed expression of *Ifnb1* (encoding IFN-β) and *Il1b* (encoding IL-1β) in BMDMs and BMDCs. As expected, *Ifnb1* was expressed only during infection with cytosolic, wild-type *L. monocytogenes* in both cell subsets (S1B Fig), where *Il1b* was induced by TLR signaling during infection with both wild-type and Δ*hly L. monocytogenes* infection (S1C Fig).

Given that *Ptgs2* expression was higher during infection with wild-type compared to Δ*hly L. monocytogenes*, we next assessed whether the increased transcript in wild-type infection led to increased COX-2 protein expression. To assess the role of cytosolic access on COX-2 protein levels, we infected BMDMs or BMDCs with wild-type or Δ*hly L. monocytogenes* and assessed COX-2 protein expression six hours later by western blot. In BMDMs, interestingly, infection with either strain of *L. monocytogenes* led to increased COX-2 protein expression (Fig 1C). In BMDCs, alternatively, Δ*hly L. monocytogenes* induced lower levels of COX-2 protein (Fig 1C). This suggests that cytosolic access is required for robust induction of COX-2 protein expression in BMDCs.

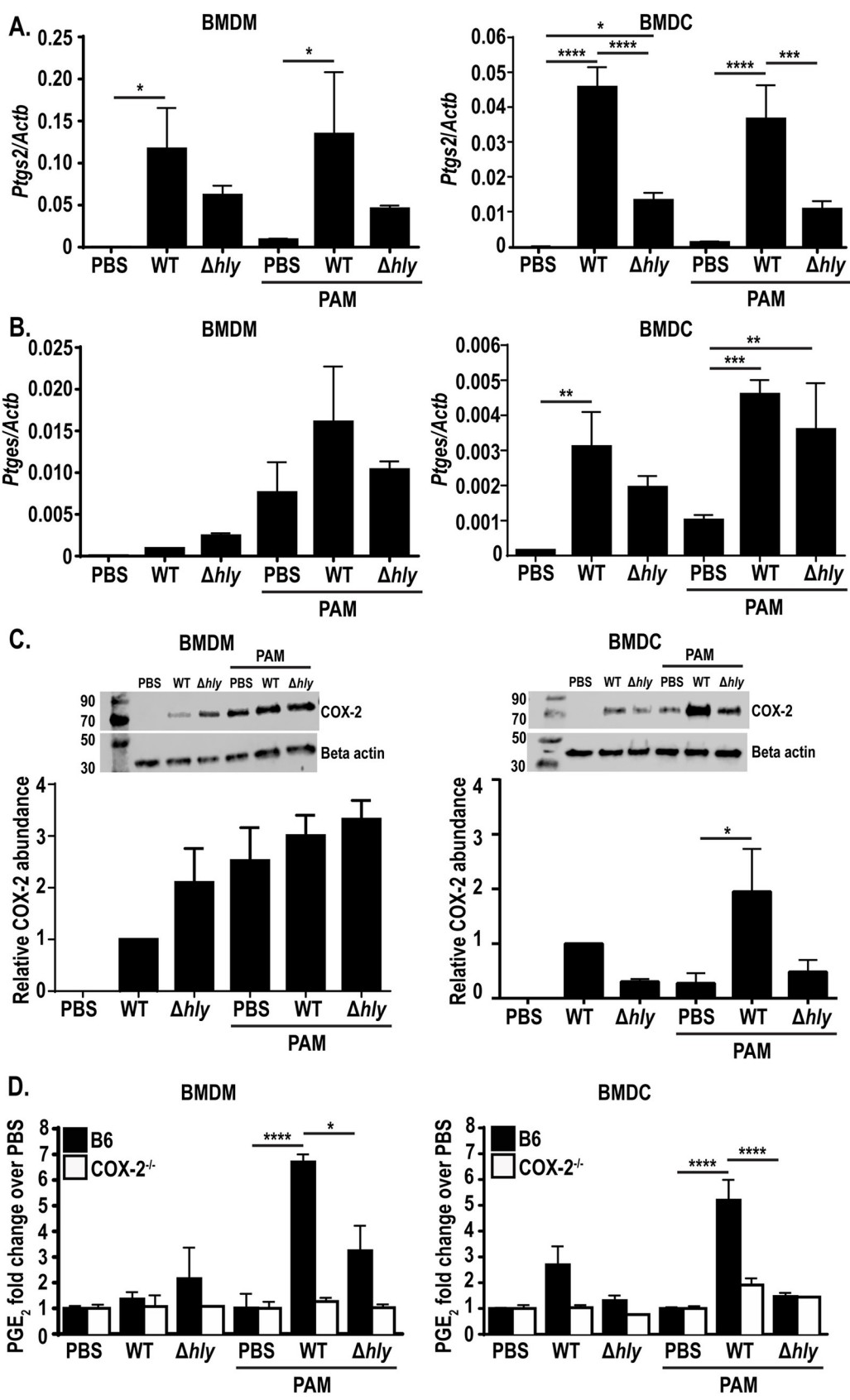

**Fig 1. PAM-primed BMDMs and BMDCs express PGE$_2$ after cytosolic infection with *L. monocytogenes*.** Wild-type or COX-2$^{-/-}$ BMDMs or BMDCs were infected with indicated strains of *L. monocytogenes* at an MOI of 10 +/- the TLR2 agonist PAM3CSK4 and assessed 6hpi for expression of *Ptgs2* (encoding COX-2) and *Ptges* (encoding mPGES-1) by qRT PCR (A-B) or COX-2 protein by western blot (C). Supernatant was harvested and assessed for PGE$_2$ by mass spectrometry (D). Mass spectrometry data was normalized to d-PGE$_2$ and fold change is relative to PBS treated controls. Data are a combination of at least two independent experiments (A,B, D, and western blot quantification), or a representative of at least two independent experiments (western blot image). Significance was determined by a one-way ANOVA with Bonferroni's correction. $^*p < 0.05$, $^{**}p < 0.01$, $^{***}p < 0.001$, $^{****}p < 0.0001$.

Infection of BMDMs and BMDCs with wild-type *L. monocytogenes* led to expression of the genes necessary for PGE$_2$ production (Fig 1A–1C). To assess whether these cells could utilize these enzymes to produce PGE$_2$, we assessed PGE$_2$ production in culture supernatant by mass spectrometry. Surprisingly, supernatant from both BMDMs and BMDCs had no detectable PGE$_2$ compared to PBS-treated controls, both during infection with wild-type or Δ*hly L. monocytogenes* (Fig 1D). This suggests that either enzyme expression was not high enough to induce detectable PGE$_2$, or there may be additional post transcriptional modifications required for enzyme activity. Analysis of PGE$_2$ from BMDMs or BMDCs deficient in COX-2 had no detectable PGE$_2$, as expected (Fig 1D).

## Primed BMDMs and BMDMs produce PGE$_2$ during cytosolic *L. monocytogenes* infection

The lack of PGE$_2$ produced by BMDMs and BMDCs in response to *L. monocytogenes* infection was surprising given the upregulation of *Ptgs2* and *Ptges* transcripts. Other innate pathways, such as the inflammasome, require a priming step in order to induce optimal activation. We hypothesized that naïve bone marrow-derived phagocytes may similarly require additional stimulation in order to produce PGE$_2$. To test this hypothesis, we treated BMDMs and BMDCs overnight with the TLR2 agonist PAM3CSK4 (PAM) before infection with wild-type and Δ*hly L. monocytogenes* and again analyzed transcript expression. PAM alone induced a small amount of expression of *Ptgs2* expression in both BMDMs and BMDCs (Fig 1A). Infection with wild-type *L. monocytogenes* led to a significant increase in expression that was less robust in Δ*hly L. monocytogenes*-infected cells, similar to the effect seen in unprimed cells (Fig 1A). *Ptges* expression, alternatively, had a larger increase in transcript expression during PAM-priming, both during wild-type and Δ*hly L. monocytogenes* infection of BMDMs and BMDCs (Fig 1B). Furthermore, PAM treatment alone induced expression of *Ptges* similar to that induced during infection in BMDMs (Fig 1B). Taken together, these data suggest that cytosolic access accentuates expression of *Ptgs2*, where TLR signaling alone is sufficient to induce *Ptges* expression. We also assessed expression of *Pla2g4a* in TLR-primed BMDMs and BMDCs and, similar to unprimed cells, saw no changes in expression (S1A Fig). Additionally, *Ifnb1* transcript was again induced during cytosolic infection, where *Il1b* expression was induced during PAM-treatment alone, as well as during infection with wild-type or Δ*hly L. monocytogenes* (S1B and S1C Fig).

As *Ptgs2* expression was also dependent on cytosolic access in primed BMDMs and BMDCs, we next assessed expression of COX-2 protein in primed cells infected with wild-type or Δ*hly L. monocytogenes* by western blot. Similar to unprimed cells, BMDMs had similar levels of COX-2 protein during infection with wild-type or Δ*hly L. monocytogenes* (Fig 1C). In BMDCs, alternatively, COX-2 protein expression was reduced during infection with Δ*hly L. monocytogenes* compared to wild-type infection (Fig 1C), again suggesting that COX-2 protein expression in BMDCs is potentiated by cytosolic access.

We hypothesized that priming BMDMs and BMDCs with PAM would stimulate the cells to produce PGE$_2$ during infection with wild-type *L. monocytogenes*. To test this hypothesis, we

assessed production of PGE$_2$ in the supernatant of primed BMDMs and BMDCs by mass spectrometry. BMDMs and BMDCs were treated overnight with PAM before infection with wild-type and $\Delta$*hly L. monocytogenes*. Six hours post-infection, cell supernatant was assessed for PGE$_2$. In contrast to unprimed BMDM and BMDCs, wild-type infection of primed cells led to a significant increase in PGE$_2$ production compared to PBS-treated controls (Fig 1D). Previous data showed *L. monocytogenes*-stimulated PGE$_2$ production in peritoneal macrophages [23,25]. Our data suggest that priming BMDMs prior to infection induces the cells to behave more like tissue resident macrophages in respect to PGE$_2$ production. Furthermore, the ability of BMDMs to produce PGE$_2$ provides a tool to efficiently study PGE$_2$ synthesis in macrophages during infection. PAM-primed COX-2 deficient BMDMs and BMDCs again led to no PGE$_2$ production, solidifying the necessity of COX-2 activity in PGE$_2$ production (Fig 1D). To characterize the kinetics of PGE$_2$ production, we analyzed PGE$_2$ production via ELISA from two to eight hours post-infection in BMDMs. During wild-type infection, PGE$_2$ was produced as early as two hours post-infection (S1D Fig). PGE$_2$ peaked at six hours post-infection, consistent with our mass spectrometry data showing elevated levels at this time point (S1D Fig). PGE$_2$ levels were maintained until at least eight hours post-infection (S1D Fig).

Importantly, maximal PGE$_2$ production in primed BMDM and BMDCs was dependent on cytosolic access, as infection with $\Delta$*hly L. monocytogenes* led to significantly reduced PGE$_2$ levels (Fig 1D). No PGE$_2$ was detected during $\Delta$*hly L. monocytogenes* infection at any earlier or later timepoints assessed, suggesting that vacuole-constrained infection does not simply alter PGE$_2$ production kinetics (S1D Fig). Furthermore, when BMDMs were infected with substantially more $\Delta$*hly L. monocytogenes* (MOI 50 compared to MOI 10 wild-type *L. monocytogenes*) there was still significantly less PGE$_2$ production at six hours-post infection (S1E Fig). Taken together, these results suggest that cytosolic *L. monocytogenes* induces robust PGE$_2$ production in TLR-primed BMDMs and BMDCs.

Given our data that cytosolic access potentiated PGE$_2$ production in BMDMs and BMDCs, we next assessed whether known cytosolic pathways influenced production of PGE$_2$. Two innate cytosolic pathways activated by *L. monocytogenes*, inflammasomes and type I IFN, can influence PGE$_2$ production in other infection models [25–27]. Accordingly, we assessed PGE$_2$ production *ex vivo* during infection with strains of *L. monocytogenes* that alter activation of these pathways. Infection of BMDMs with $\Delta$*tetR L. monocytogenes* (a strain that hyperactivates type I IFN) [28] or $\Delta$*mdrMTAC* (a strain that induces less type I IFN) [29] did not alter PGE$_2$ production (S1F Fig). In contrast, infection with a strain that hyperactivates the inflammasome, Lm-pyro, led to a small but reproducible decrease in PGE$_2$ production (S1F Fig), suggesting that inflammasome activation may influence PGE$_2$ production in response to *L. monocytogenes* infection.

Lastly, we also sought to understand whether PGE$_2$ specifically was being induced, or if there was a more broad increase eicosanoid production. To test the hypothesis that *L. monocytogenes* induces production of other eicosanoids, we analyzed production of prostaglandin D$_2$ (PGD$_2$), thromboxane B$_2$ (TXB$_2$), and leukotriene B$_4$ (LTB$_4$). However, we saw no changes production of these eicosanoids by wild-type *L. monocytogenes* (S2A and S2B Fig). This interesting observation suggests that macrophages and dendritic cells preferentially induce PGE$_2$ in response to infection with cytosolic *L. monocytogenes*.

## Cytosolic access is required for PGE$_2$ production *in vivo*

Production of PGE$_2$ by TLR-primed BMDMs and BMDCs *ex vivo* is potentiated by cytosolic access. To assess whether *L. monocytogenes* induces PGE$_2$ in a cytosol-dependent manner *in vivo*, we infected mice intravenously with wild-type and $\Delta$*hly L. monocytogenes* and assessed

PGE$_2$ levels in the spleen twelve hours post-infection, previously defined as the peak PGE$_2$ response to infection [16]. Wild-type *L. monocytogenes* led to an eight-fold increase in PGE$_2$ (Fig 2A). Infection with Δ*hly L. monocytogenes* strikingly showed no increase in PGE$_2$ over mock-immunized controls (Fig 2A). To ensure that the reduced PGE$_2$ production was not due to differences in bacterial burdens, mice were infected at a dose of wild-type ($10^5$ bacteria) and Δ*hly L. monocytogenes* ($10^7$ bacteria) that led to comparable burdens (Fig 2B). This shows that the absence of PGE$_2$ in Δ*hly L. monocytogenes*-infected mice is not just due to reduced bacterial burdens. Taken together, these data highlight that cytosolic access is necessary for *in vivo* induction of PGE$_2$.

Previous data demonstrated that infection with Δ*hly L. monocytogenes* fails to induce robust cell-mediated immunity [11,12]. Furthermore, coinfection of Δ*hly L. monocytogenes* with wild-type *L. monocytogenes* suppresses T-cell priming associated with wild-type infection, demonstrating that infection with vacuole-constrained *L. monocytogenes* induces an immuno-suppressive environment [11]. We hypothesized that infection with Δ*hly L. monocytogenes* may actively inhibit PGE$_2$ production, contributing to impaired T-cell responses. To address this hypothesis, we infected mice with wild-type *L. monocytogenes*, Δ*hly L. monocytogenes*, or a coinfection of both strains and assessed PGE$_2$ levels twelve hours post-infection. Wild-type *L. monocytogenes* infection induced PGE$_2$ whereas Δ*hly L. monocytogenes* failed to induce production, as expected (Fig 2C). Strikingly, coinfection of the two strains led to little to no PGE$_2$ production (Fig 2C) suggesting that vacuole-constrained strains of *L. monocytogenes* actively impair production of PGE$_2$. Active inhibition of PGE$_2$ production, combined with the previously described impacts of TLR-dependent IL-10[11], could be one of multiple reasons why Δ*hly L. monocytogenes* fails to induce protective immunity.

## CD11c$^+$ and Lyz2$^+$ cells produce PGE$_2$ during *L. monocytogenes* infection *in vivo*

Our data identified PGE$_2$ production by macrophages and dendritic cells *ex vivo* (Fig 1D). Furthermore, previous groups have reported that macrophage and dendritic cell subsets are heavily infected early during *in vivo* infection, a timepoint where we have previously detected increases in splenic PGE$_2$ [16,22]. From these data, we next hypothesized that macrophages and/or dendritic cells were responsible for producing PGE$_2$ *in vivo* that is necessary for optimal T-cell priming. Accordingly, we hypothesized that deletion of PGE$_2$ production in these cells specifically would 1) reduce the splenic levels of PGE$_2$ and 2) impair the robust CD8$^+$ T-cell responses normally induced by *L. monocytogenes*. To test these hypotheses, we generated mice deficient in COX-2 selectively in CD11c$^+$ cells or Lyz2$^+$ cells using the cre/lox system. Mice containing *loxP* sites flanking the COX-2-encoding gene (COX-2$^{fl/fl}$) were crossed with mice expressing the cre recombinase under the CD11c (CD11c-cre$^+$) or Lyz2 (Lyz2-cre$^+$) promoters. COX-2$^{fl/fl}$ CD11c-cre$^+$ and COX-2$^{fl/fl}$ Lyz2-cre$^+$ mice were immunized with $10^7$ CFU of a live-attenuated, vaccine strain of *L. monocytogenes* (LADD *L. monocytogenes*) currently used in clinical trials as a cancer therapy platform [30]. The LADD strain is deficient in two major virulence genes, *actA* and *inlB*, but retains immunogenicity, making it safe for clinical use [30,31]. The LADD vaccine strain was used here to enable analysis of T-cell responses in floxed mice as discussed below and induces similar levels of PGE$_2$ [16].

We first hypothesized that immunization of COX-2$^{fl/fl}$ CD11c-cre$^+$ and/or COX-2$^{fl/fl}$ Lyz2-cre$^+$ mice with LADD *L. monocytogenes* would lead to lower levels of splenic PGE$_2$ compared to control mice. Immunization of COX-2$^{fl/fl}$ CD11c-cre$^+$ and COX-2$^{fl/fl}$ Lyz2-cre$^+$ mice each showed reduced levels of PGE$_2$ production, leading to only 60% of the PGE$_2$ induced during immunization of control mice (Fig 3A). However, deletion of COX-2 in either CD11c$^+$ or

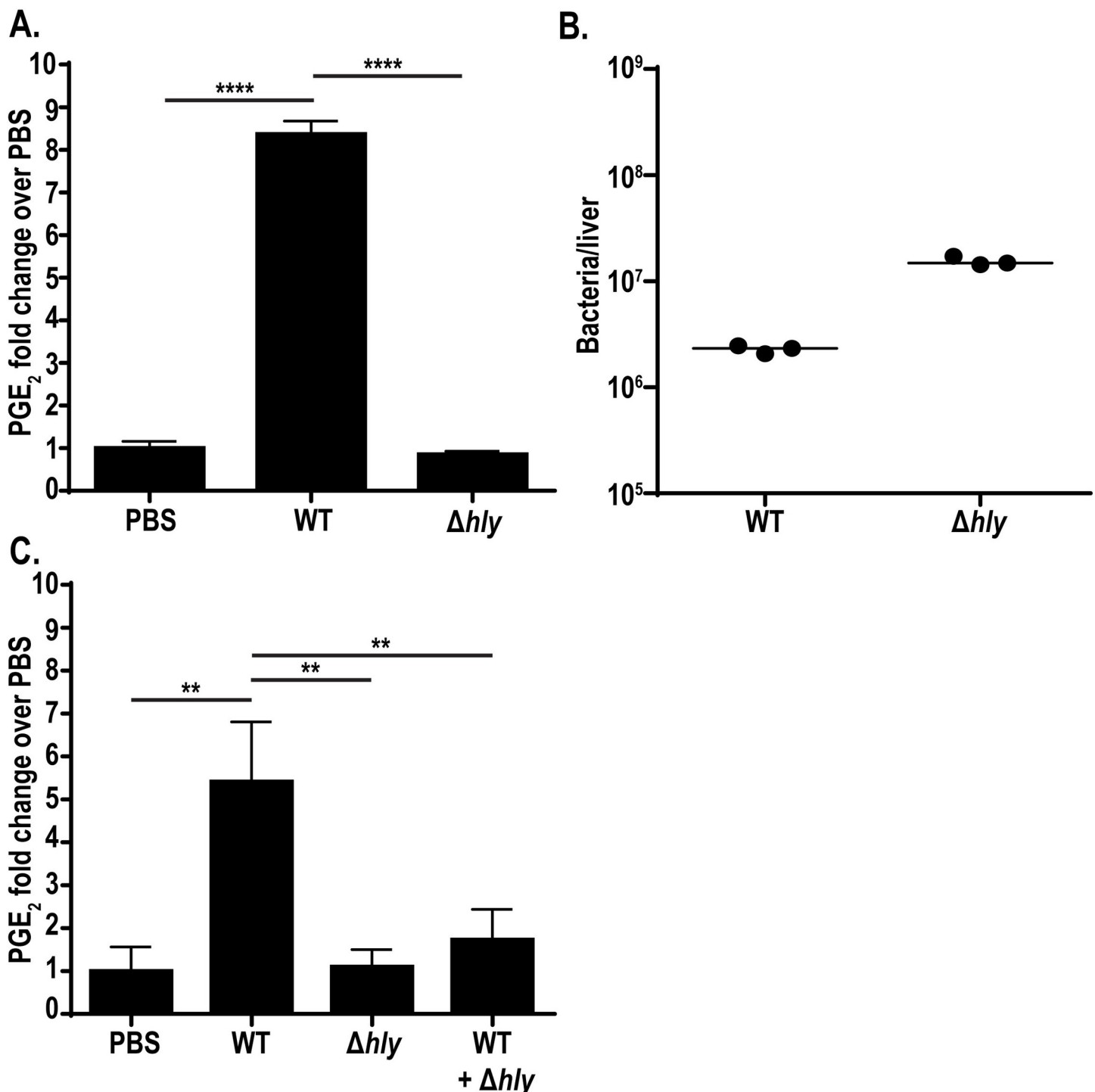

**Fig 2. Cytosolic access is necessary for *L. monocytogenes*-stimulated PGE$_2$ production *in vivo*.** C57BL/6 mice were infected with 10$^5$ wild-type or 10$^7$ Δ*hly L. monocytogenes*. 12hpi spleens were harvested for eicosanoid extraction and mass spectrometry (A) and livers were harvested for bacterial burdens (B). C57BL/6 mice were infected with 10$^5$ wild-type, 10$^7$ Δ*hly L. monocytogenes*, or a combination of both strains. 12hpi spleens were harvested for eicosanoid extraction and mass spectrometry (C). Data are representative of two independent experiments. Mass spectrometry data was normalized to d-PGE$_2$ levels and fold change is compared to PBS controls. Significance was determined by a one-way ANOVA with Bonferroni's correction. ** $p < 0.01$, **** $p < 0.0001$.

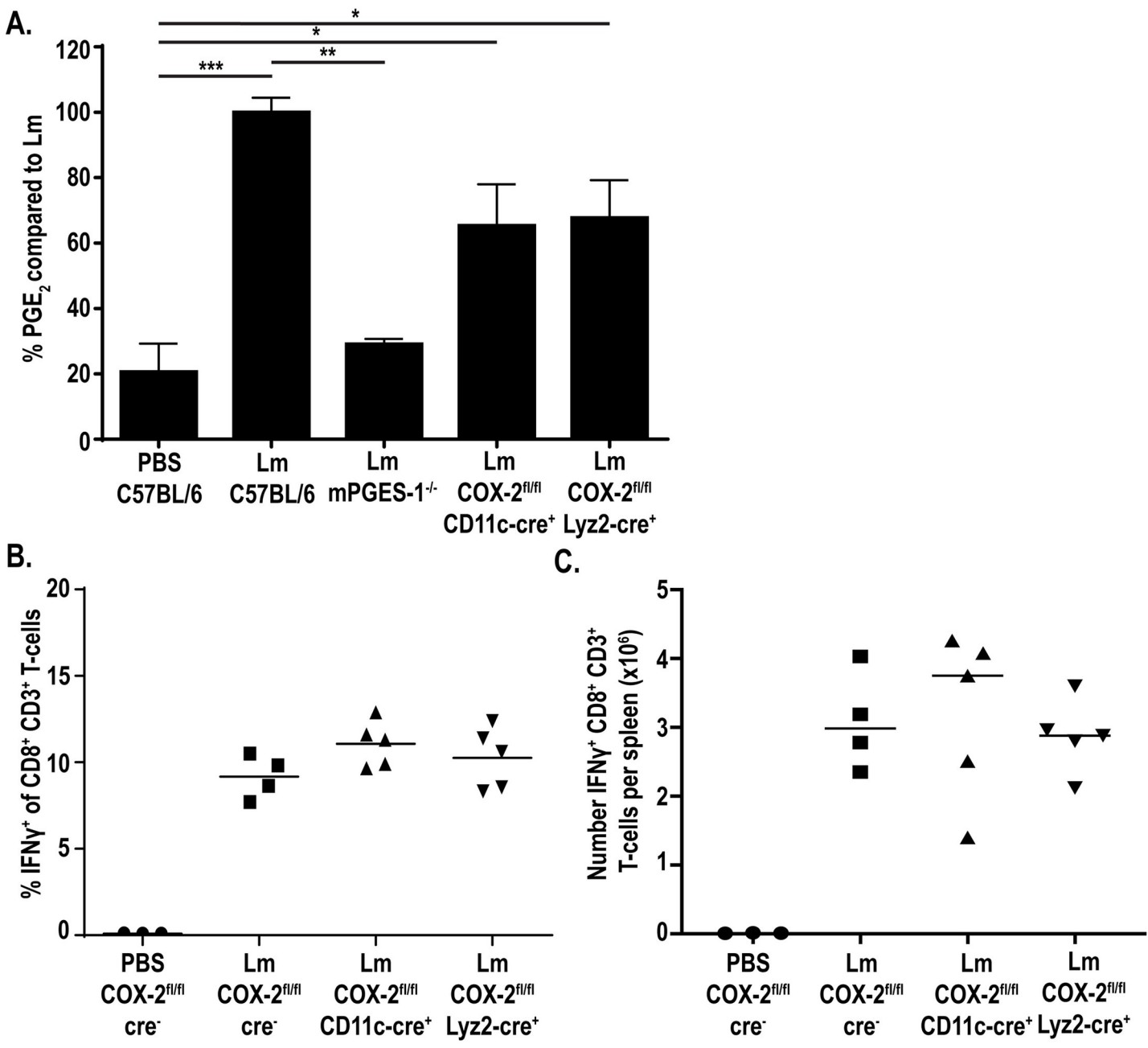

**Fig 3. CD11c⁺ and Lyz2⁺ cells contribute to PGE₂ production *in vivo*.** Indicated strains of mice were infected with $10^7$ LADD *L. monocytogenes*. 12hpi spleens were harvested and assessed for PGE₂ by mass spectrometry. Data was normalized to d-PGE₂ levels and percent change is compared to *L. monocytogenes*-infected controls (A). Indicated strains of mice were infected with $10^7$ LADD *L. monocytogenes*. 7dpi splenocytes were examined for B8R-specific CD8⁺ T-cell responses. %IFNγ (B) or number IFNγ (C) per spleen was assessed. Data shown are representative of two independent experiments. Significance was determined by a one-way ANOVA with Bonferroni's correction (A). $^*p < 0.05$, $^{**}p < 0.01$, $^{***}p < 0.001$.

Lyz2⁺ cells did not fully abrogate production of PGE₂ as observed in mice globally deficient in mPGES-1 (mPGES-1⁻ᐟ⁻) (Fig 3A). This suggests that CD11c⁺ and Lyz2⁺ cells each contribute to PGE₂ production and that deletion of COX-2 in either is not sufficient to completely prevent PGE₂ production. We also assessed PGE₂ levels in mice deficient in COX-2 selectively in T-cells and PGE₂ production was still observed (S3 Fig). As T-cells are not known to be

infected by *L. monocytogenes*, this is consistent with our hypothesis suggesting PGE$_2$ production specifically from infected cell subsets.

PGE$_2$ is critical for generating optimal T-cell responses in response to *L. monocytogenes*, as immunization of mPGES-1-deficient mice or treatment of mice with a COX-2-specific pharmacological inhibitor leads to impaired CD8$^+$ T-cell responses [16]. We next hypothesized that the decreased PGE$_2$ production in the COX-2$^{fl/fl}$ CD11c-cre$^+$ or COX-2$^{fl/fl}$ Lyz2-cre$^+$ mice would be sufficient to similarly impair CD8$^+$ T-cell responses. We specifically assessed the impact on CD8$^+$ T-cells as these are the predominate cells contributing to protective immunity to *L. monocytogenes* [1,2] and are cells of interest in *L. monocytogenes*-based cancer immunotherapeutics [4]. To test this hypothesis, we immunized mice with 10$^7$ LADD *L. monocytogenes* expressing the model antigens B8R and OVA. Seven days after immunization, splenocytes were isolated, stimulated with B8R or OVA, and production of IFNγ was assessed by flow cytometry. Despite decreased PGE$_2$ production in these mice, T-cell responses were not affected both in percent IFNγ$^+$ as well as number of IFNγ$^+$ T-cells per spleen (Figs 3B, 3C, S4A and S4B). Similarly, the number of antigen-specific T-cells measured by B8R tetramer was unchanged in these mice compared to control mice (S4C Fig). This suggests that the PGE$_2$ remaining in these mice was sufficient to prime productive T-cell responses. Due to its short *in vivo* half-life, PGE$_2$ asserts its effects locally [32]. It is possible that while global splenic PGE$_2$ levels are decreased, the local concentrations of PGE$_2$ are sufficient to prime T-cell responses. Taken together, these data suggest that although Lyz2$^+$ and CD11c$^+$ cells contribute to production of PGE$_2$ during *L. monocytogenes* infection, PGE$_2$ production by either cell subset alone these cells is not necessary for T-cell priming, as CD8$^+$ T-cell responses are not impacted by loss of PGE$_2$ production in either subset.

## Deletion of COX-2 in both Lyz2$^+$ and CD11c$^+$ cells further reduces splenic PGE$_2$ levels

Our data showed that single deletions of COX-2 in CD11c$^+$ or Lyz2$^+$ cells reduced PGE$_2$, but not to baseline values. We next hypothesized that PGE$_2$ production by either of these subsets individually was sufficient for T-cell priming and that to observe impaired T-cell responses we would have to eliminate PGE$_2$ production in both CD11c$^+$ and Lyz2$^+$ cells. To do this, we crossed the COX-2$^{fl/fl}$ CD11c-cre$^+$ and COX-2$^{fl/fl}$ Lyz2-cre$^+$ mice, leading to mice with a COX-2 deletion in both cell subsets (COX-2$^{fl/fl}$ CD11c-cre$^+$ Lyz2-cre$^+$). We assessed the ability of these mice to produce PGE$_2$ by mass spectrometry and found that PGE$_2$ was further reduced, with about 40% the amount PGE$_2$ produced compared to immunized control mice (Fig 4A). This suggests that CD11c$^+$ and Lyz2$^+$ cells combined to produce the majority of PGE$_2$ during immunization with *L. monocytogenes*.

Due to further reduced PGE$_2$ production in our mice deficient in COX-2 in both CD11c$^+$ and Lyz2$^+$ cells, we next assessed CD8$^+$ T-cell responses in these mice. Mice again were immunized with 10$^7$ vaccine strain of *L. monocytogenes* expressing the model antigens B8R and OVA and assessed for IFNγ production seven days later. Despite diminished PGE$_2$ production, CD8$^+$ T-cell responses remained intact in the double COX-2 deficient mice, both in percent and number (Figs 4B, 4C, S5A and S5B). Similarly, antigen-specific T-cells measured by B8R tetramer were also unchanged compared to wild-type controls (S5C Fig). This suggest that even the small amount of PGE$_2$ produced locally is sufficient to drive CD8$^+$ T-cell responses.

## Depletion of phagocytes eliminates PGE$_2$ production *in vivo*

Our *ex vivo* data highlighted the ability of BMDMs and BMDCs to produce PGE$_2$ in response to cytosolic *L. monocytogenes*. However, deletion of COX-2 in Lyz2$^+$ and CD11c$^+$ cells did not

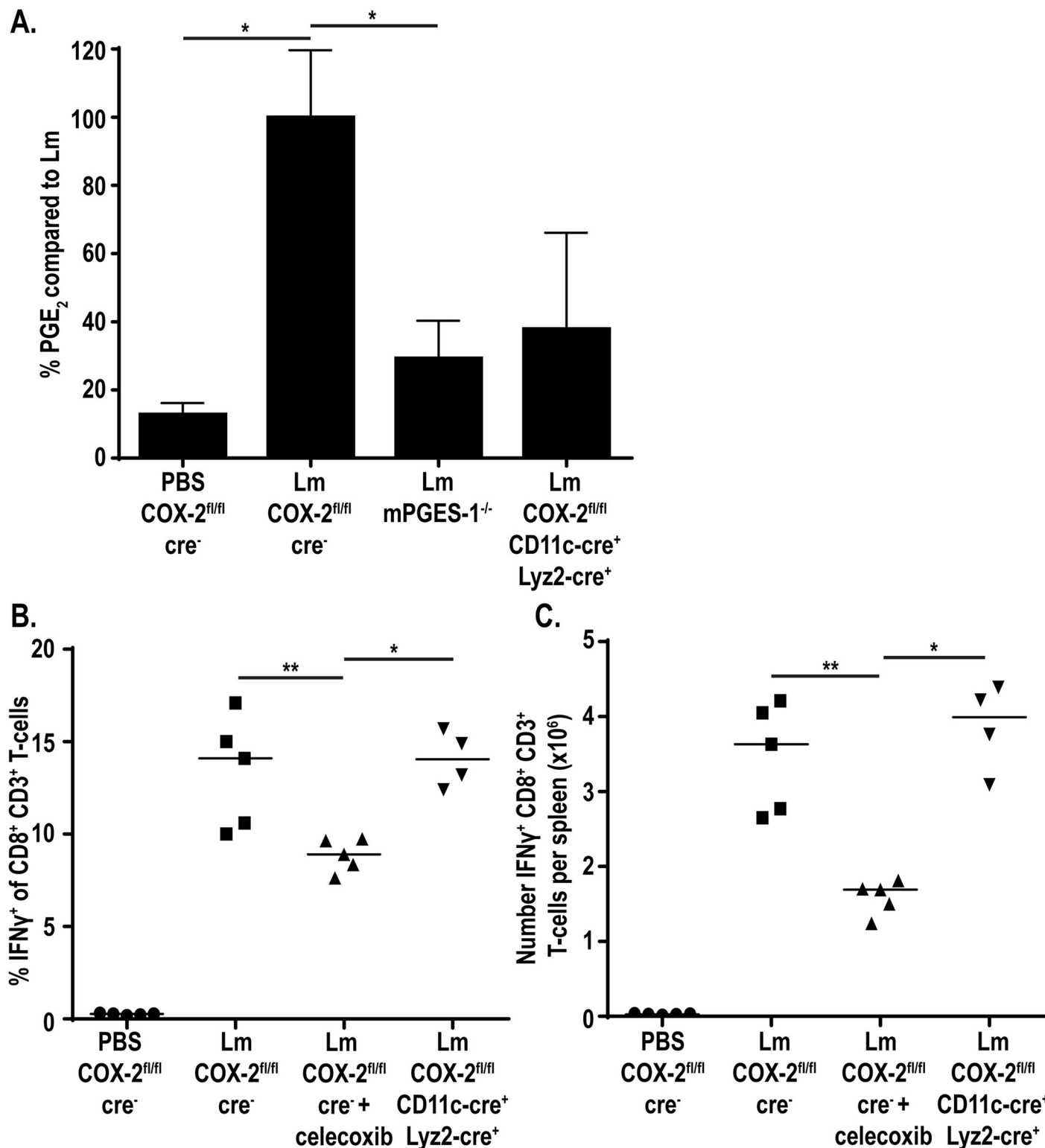

**Fig 4. Deletion of COX-2 in both CD11c⁺ and Lyz2⁺ cells further reduces PGE₂ production.** Indicated strains of mice were infected with $10^7$ LADD *L. monocytogenes*. 12hpi spleens were harvested and assessed for PGE₂ by mass spectrometry. Data was normalized to d-PGE₂ levels and percent change is compared to *L. monocytogenes*-infected controls (A). Indicated strains of mice were infected with $10^7$ LADD *L. monocytogenes*. 7dpi splenocytes were examined for B8R-specific CD8⁺ T-cell responses. %IFNγ (B) or number IFNγ (C) per spleen was assessed. Data shown are representative of two independent experiments of 3–5 mice per group. Significance was determined by a one-way ANOVA with Bonferroni's correction (A) or a Mann-Whitney *U* test (B-C). *$p < 0.05$, **$p < 0.01$.

completely abrogate PGE$_2$ production *in vivo*. These data led us to hypothesize that other phagocytic cell subsets not effectively targeted by these cre-drivers may be producing the residual PGE$_2$, such as marginal zone macrophages (MZMs), metallophilic macrophages, or other CD11b$^+$ cells more broadly [33,34]. To test this hypothesis, we utilized short-term clodronate liposomes to rapidly deplete phagocyte populations in the spleen. Mice were depleted with clodronate liposomes 24 hours prior to immunization with *L. monocytogenes*[35]. Twelve hours post-immunization, spleens were harvested and assessed for PGE$_2$ by mass spectrometry. Additionally, splenocytes were assessed for CD11b$^+$ and CD11c$^+$ populations by flow cytometry to confirm clodronate efficacy. Clodronate treatment led to significantly fewer CD11b$^+$ cells and a trend for decreased CD11c$^+$ cells (S6A and S6B Fig). Treatment of mice with clodronate prior to infection with *L. monocytogenes* completely eliminated PGE$_2$ production compared to infected control mice (Fig 5A). Importantly, bacterial burdens were equivalent between clodronate and mock-treated mice (Fig 5B). Pretreatment with a control empty liposome, encapsome, actually increased PGE$_2$ production compared to infected control mice, potentially due to increased bacterial burdens (Fig 5A and 5B). To ensure that the abrogated PGE$_2$ was due to phagocyte depletion and not due to overall increased levels of cell death associated with clodronate-mediated depletion, additional mice were treated with a B cell-depleting anti-CD20 antibody 24 hours prior to infection with *L. monocytogenes*. Treatment with anti-CD20 resulted in depletion of ~50% of splenic B-cells, yet did not alter PGE$_2$ production compared to mice treated with an isotype control (S6C and S6D Fig). These data suggest that the loss of PGE$_2$ production in the context of clodronate treatment was specific to depletion of phagocytes and that generic immune cell depletion and associated cell death does not prevent PGE$_2$ production. Taken together, these data demonstrate that phagocytic cell populations are critical for PGE$_2$ production *in vivo* following *L. monocytogenes* immunization.

Loss of antigen presenting cells through clodronate treatment leads to impaired CD8$^+$ T-cell activation due to loss of antigen presenting cells, making analysis of CD8$^+$ T-cell responses in this model not informative [36,37]. Given this, we alternatively assessed the possibility that other phagocytic cells targeted by clodronate, but not the COX-2$^{fl/fl}$ Lyz2-cre$^+$, could contribute to PGE$_2$ production. Complete elimination of PGE$_2$ production with clodronate treatment suggested that the residual PGE$_2$ in the COX-2$^{fl/fl}$ CD11c-cre$^+$ Lyz2-cre$^+$ mice was due to a phagocytic cell that was not effectively targeted in these mice. Previous data showed that although the Lyz2-cre used in this study is highly efficient at deletion of *loxP* flanked genes in some macrophage subsets, it is only minimally successful at deleting genes of interest in other subsets, such as MZMs [33]. MZMs, characterized by expression of MARCO, are heavily infected early in *L. monocytogenes* infection [22]. We hypothesized that the residual PGE$_2$ we detected in the COX-2$^{fl/fl}$ CD11c-cre$^+$ Lyz2-cre$^+$ mice may be due to inefficient deletion in macrophage subsets such as these. To assess the role of MZMs in PGE$_2$ production, we assessed expression of COX-2 by immunohistochemistry. Mice were immunized with 10$^7$ vaccine strain of *L. monocytogenes* and spleens were harvested three and ten hours later. Spleen cryosections were then stained for *L. monocytogenes*, COX-2, and MARCO. Uninfected mice had COX-2 staining in the periarteriolar lymphoid sheath (PALS) with little expression in the marginal zone (MZ) (Fig 5C and 5D). As early as three hours post-immunization COX-2 staining was observed in the MZ, with approximately 50% of COX-2 colocalizing with MARCO$^+$ cells (Fig 5C and 5D). Expression of COX-2 in the MZ was maintained at 10hpi, again showing approximately 50% colocalization with MARCO (Fig 5C and 5D). Furthermore, *L. monocytogenes* colocalized with COX-2 and MARCO expressing cells, suggesting that infected MZMs may be producing PGE$_2$ (Fig 5D). Expression of COX-2 suggests that MZMs, or other non-CD11c/Lyz2 expressing phagocytes within the marginal zone, could be capable of producing PGE$_2$ *in vivo* and may be contributing to the PGE$_2$ remaining in the COX-2$^{fl/fl}$

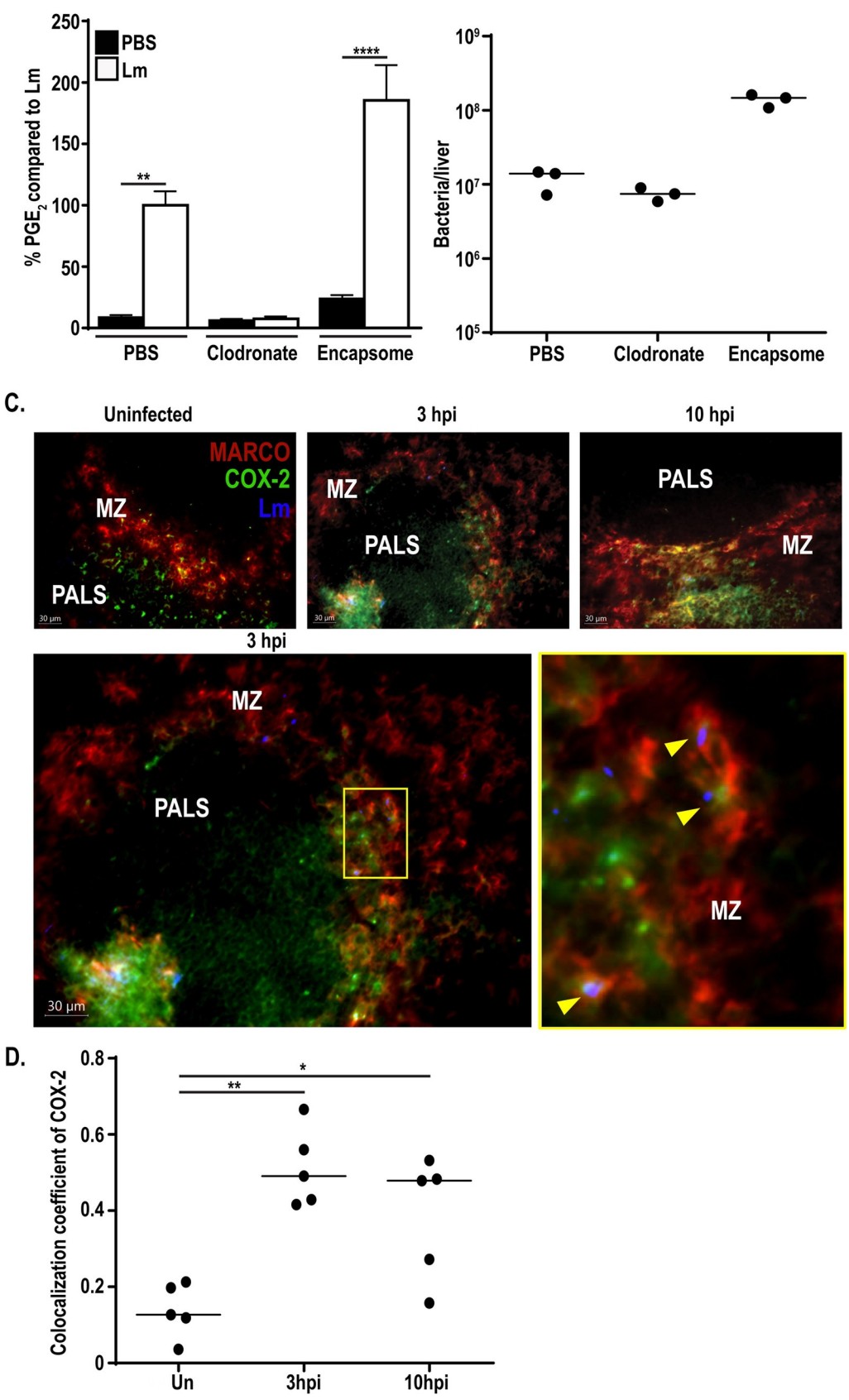

**C.**

| Uninfected | 3 hpi | 10 hpi |

MARCO
COX-2
Lm

MZ
PALS

MZ
PALS

PALS
MZ

3 hpi

MZ
PALS

MZ

**D.**

**Fig 5. Phagocyte depletion eliminates PGE$_2$ production.** C57BL/6 mice were dosed with 200μL clodronate, liposome control (encapsome), or PBS 24 hours prior to immunization with $10^7$ LADD *L. monocytogenes*. 12hpi spleens were harvested and assessed for PGE$_2$ by mass spectrometry. Data was normalized to d-PGE$_2$ levels and percent change is compared to *L. monocytogenes*-infected controls (A). Livers were harvested concurrently and assessed for bacterial burdens (B). C57BL/6 mice were immunized with $10^7$ LADD *L. monocytogenes*. 3 and 10hpi spleens were harvested, cryosections were cut, and sections were stained for *L. monocytogenes* (Lm), COX-2, and MARCO (C). Yellow arrows represent colocalization of *L. monocytogenes*, COX-2, and MARCO (C). Colocalization coefficients (Pearson's correlation) of COX-2 and MARCO were calculated (D). Data shown are representative of at least two independent experiments. Significance was determined by a one-way ANOVA with Bonferroni's correction (A) or a Mann-Whitney *U* test (D). $^*p < 0.05$, $^{**}p < 0.01$, $^{****}p < 0.0001$.

CD11c-cre$^+$ Lyz2-cre$^+$ mice. To test the hypothesis that MZMs could contribute to the residual PGE$_2$ observed in the double transgenic mice, we assessed expression of COX-2 by immuno-histochemistry in the COX-2$^{fl/fl}$ CD11c-cre$^+$ Lyz2-cre$^+$ mice. The mice were infected with $10^7$ LADD *L. monocytogenes* or mock-infected with PBS and spleens were harvested 10 hours later. COX-2 expression in the PALS was completely eliminated, confirming the efficacy of the Lyz-cre and CD11-cre in deletion of COX-2 (S6E Fig). Importantly, COX-2 expression was still detected in the MZ and colocalized with MARCO, supporting the hypothesis that MZMs may contribute to the residual PGE$_2$ (S6E Fig). Taken together, our data suggest that multiple myeloid derived subsets can contribute to PGE$_2$ production, including Lyz2$^+$ cells, CD11c$^+$ cells, and possibly MZMs. Complete reductions in PGE$_2$ by depletion of phagocytic cells such as these with clodronate treatment is consistent with our data showing that PGE$_2$ is produced from phagocytic cells infected with cytosolic *L. monocytogenes*.

## Discussion

Cytosolic access is required to effectively generate cell-mediated immunity to *L. monocytogenes*[10–12]. Decades of work has focused on understanding the cytosol-dependent processes necessary for T-cell priming, a topic that has gained interest recently due to use of *L. monocytogenes* as a cancer immunotherapy platform. Our data suggest that one reason cytosolic access is important may be to facilitate phagocyte production of PGE$_2$, an eicosanoid required to generate optimal CD8$^+$ T-cell responses [16]. We showed that PGE$_2$ is produced by BMDMs and BMDCs *ex vivo*. Importantly, this pathway is potentiated by cytosolic access, as vacuole-constrained *L. monocytogenes* induce lower production of PGE$_2$. Furthermore, infection of mice with a vacuole-constrained *L. monocytogenes* strain led to no increase of PGE$_2$ over mock immunized controls. Lastly, we showed that Lyz2$^+$ and CD11c$^+$ cells contribute to PGE$_2$ production *in vivo* as deletion of COX-2 in these subsets led to decreased PGE$_2$ levels, however other clodronate-sensitive phagocyte populations also contribute to PGE$_2$ production following *L. monocytogenes* immunization. This work leads to many new questions including how cytosolic *L. monocytogenes* activates this pathway, how immune cells discriminate which eicosanoid to produce in response to infection, how even small concentrations of PGE$_2$ still lead to productive PGE$_2$ responses, and how PGE$_2$ facilitates optimal T-cell priming.

One intriguing hypothesis is that PGE$_2$ synthesis during *L. monocytogenes* infection is driven by an innate cytosolic sensor. *L. monocytogenes* elicits a number of innate pathways that could contribute to differential activation of the PGE$_2$-synthesis pathway. One possibility is that induction of type I IFN influences PGE$_2$ production. Type I IFN can be induced cytosoli-cally by *L. monocytogenes* through recognition of cyclic diadenosine monophosphate (c-di-AMP). Upon entry into the cytosol, *L. monocytogenes* secretes c-di-AMP through multidrug resistance transporters [28,38] where it is recognized by either the reductase controlling NF-κB (RECON) [39] or stimulator of IFN genes (STING) [40,41]. STING activation leads to type I interferon induction [40,41], and was originally hypothesized to be critical for T-cell

responses. Paradoxically, however, type I IFN inhibits cell-mediated immunity to *L. monocytogenes* [14]. Interestingly, there has been well documented crosstalk between the $PGE_2$ and type I IFN pathways during infections with other pathogens such as influenza and *M. tuberculosis* [26,27]. In the context of influenza, Coulombe et al. showed that infection led to upregulation of $PGE_2$ and a subsequent decrease in type I IFN [26]. In contrast to *L. monocytogenes*, type I IFN is important in generating cell-mediated immune responses to influenza. Accordingly, diminished type I IFN due to increased $PGE_2$ reduces both acute and protective immunity during influenza infection. On the other hand, Mayer-Barber et al. recently showed that inhibition of type I IFN during *M. tuberculosis* infection led to an increased level of $PGE_2$ in an IL-1-dependent manner [27]. This correlated with better bacterial control. Due to crosstalk between these two pathways, it seems possible that recognition of c-di-AMP and subsequently upregulation of type I IFN may also be playing a role in $PGE_2$ production during *L. monocytogenes* immunization. Here, we show a preliminary analysis of the role of type I IFN on $PGE_2$ production during *L. monocytogenes* infection. In an *ex vivo* model, strains of *L. monocytogenes* that alter type I IFN levels did not change $PGE_2$ production, suggesting that perhaps type I IFN does not regulate $PGE_2$. Despite these *ex vivo* findings, future *in vivo* analysis of $PGE_2$ levels in mice deficient in STING or the type I IFN receptor (IFNAR) is necessary to conclusively define links between these two pathways. Should type I IFN negatively influence $PGE_2$ production *in vivo*, use of *L. monocytogenes* strains that have reduced secretion of c-di-AMP and subsequently less type I IFN could be an avenue of further research for immunotherapeutic platforms.

Another cytosolic pathway that may influence $PGE_2$ levels during *L. monocytogenes* infection is the inflammasome. Inflammasomes are multiprotein complexes that recognize a wide range of pathogen associated molecular patterns [42–44]. Wild-type *L. monocytogenes* infection leads to a small amount of inflammasome activation, largely through the absent in melanoma 2 (AIM2) inflammasome [45]. The AIM2 inflammasome recognizes cytosolic DNA that is released during bacteriolysis within the cytosol [45–47]. Originally, inflammasomes such as AIM2 were known to have two major downstream effects, the release of proinflammatory cytokines IL1β/IL-18 and the induction of a lytic form of cell death, pyroptosis, characterized by formation of membrane pores by the protein Gasdermin D [48–50]. Seminal work by von Moltke et al. introduced a new downstream effect, the activation of an eicosanoid storm, including $PGE_2$ [25]. This work, as well as supporting recent work, showed elevated levels of $PGE_2$ after inflammasome activation [25,51]. One possible hypothesis stemming from this work is that induction of membrane pores during pyroptosis leads to calcium influx, activating cPLA2 and releasing arachidonic acid from the membrane. This model would suggest that use of mice deficient in caspase-1 or Gasdermin D would lead to lower levels of $PGE_2$ production. Intriguingly, our *ex vivo* data suggest the opposite during *L. monocytogenes* infection. During infection with a strain of *L. monocytogenes* that hyperactivates the inflammasome we found a slight, but reproducibly significant decrease in $PGE_2$ production. We hypothesize that this is likely due to increased pyroptosis in host cells, leading to a lower number of viable cells able to produce $PGE_2$. Alternatively, the diminished $PGE_2$ could be due to crosstalk between these two pathways. This finding is consistent with other data showing that hyperactivation of the inflammasome leads to impaired cell-mediated immune responses [6,52], leading to the possibility that one reason for impaired responses in the context of robust inflammasome activation may be due to lower $PGE_2$. Further analysis of $PGE_2$ levels *in vivo* during hyper-inflammasome activation and/or in mice lacking inflammasome components will be important to elucidate these details. The role of inflammasomes as well as type I IFN are intriguing avenues to understand signaling pathways driving $PGE_2$ production during *L. monocytogenes* infection.

It is also possible that production of PGE$_2$ by *L. monocytogenes* is independent of known cytosolic pathways. Identification of other unknown censors could be accomplished by assessing PGE$_2$ levels in response to different *L. monocytogenes* mutants. Mutant strains of *L. monocytogenes* that differentially induce PGE$_2$ could provide insight as to the cytosolic censors involved. One additional hypothesis is that PGE$_2$-production is independent of a cytosolic sensor completely and instead is driven by LLO-mediated pore formation. Though LLO is tightly regulated transcriptionally, translationally, and posttranslationally to be most active in the vacuole, a small amount of LLO may remain active in the cytosol of cells [53–55]. It is possible that this small amount of LLO induces pore formation in the cell membrane and allows calcium influx, subsequently activating cPLA2. Use of strains that further restrict LLO production in the cytosol, such as new strains that excise *hly* once *L. monocytogenes* has entered the cytosol [56], could help assess the role of LLO-mediated pores on PGE$_2$ production.

In addition, our data show an interesting phenotype where BMDCs and PAM-primed BMDMs selectively produce PGE$_2$ in response to *L. monocytogenes* infection rather than a global increase in eicosanoid production. Analysis of the eicosanoid milieu in cell culture supernatant show an increase in PGE$_2$ production during wild-type *L. monocytogenes* infection, but little to no changes in other eicosanoids such as PGD$_2$, TXB$_2$, or LTB$_4$. This raises the question of how a cell determines which eicosanoid to produce in response to different stimuli. The eicosanoid produced in different conditions is dependent on terminal synthases [21]. Therefore, the expression and activity of these synthases determine the resulting eicosanoid milieu. Multiple factors impact expression of different synthases including cytokines, hormones, and microbial products [57]. For example, expression of mPGES-1 can be induced by LPS and prostaglandin D$_2$ synthases, though less well understood, can be upregulated by glucocorticoids [58–60]. Conversely, anti-inflammatory cytokines such as IL-10 can inhibit expression of mPGES-1 [61]. Given that vacuole constrained *L. monocytogenes* have previously been demonstrated to induce high levels of IL-10 leading to inhibition of T-cell priming, it is interesting to speculate that the inhibition of PGE$_2$ production observed in our co-infection experiments could be tied to this previously observed induction of IL-10. Use of other *L. monocytogenes* strains that differentially activate cytokines or are deficient in different microbial PAMPs could be informative as to which signal specifically leads to enhanced levels of mPGES-1 transcript. In addition, activity of each synthase also may dictate which eicosanoids are produced [57]. Terminal synthase activity can be modulated by posttranslational modification (such as phosphorylation) as well as presence of cofactors (such as ATP and glutathione) [57]. Depletion of essential cofactors during metabolic or oxidative stress could influence the induced inflammatory milieu [57]. The post transcriptional regulation highlights the necessity of assessing endpoint eicosanoid production rather than simply transcript or protein levels, as these other factors influencing activity can alter which eicosanoids ultimately are produced. This is particularly true in our data, as despite seeing upregulation of mPGES-1 transcript during infection of unprimed BMDMs and BMDCs, we failed to see PGE$_2$ production. This suggests that perhaps some additional modification is necessary to induce mPGES-1 activity during *L. monocytogenes* infection. Similarly, our data suggest that cytosolic access is not required for COX-2 translation in primed BMDMs, but is required to fully induce PGE$_2$ production. This again suggests that further modification of COX-2 may be important for optimal activity. In contrast, cytosolic access is required for both COX-2 protein translation and PGE$_2$ production in BMDCs. Perhaps PGE$_2$ production in BMDCs is driven by COX-2 protein expression, whereas in BMDMs it is driven by post translational modifications. Further analysis on the role of phosphorylation and cofactor availability will help elucidate these details of regulation.

Another pressing question generated from this work is how productive T-cell responses were induced in mice deficient in COX-2 in both CD11c$^+$ and Lyz2$^+$ cells despite reduced PGE$_2$ levels. Here, we show that these mice produce substantially reduced PGE$_2$, yet still induce wild-type CD8$^+$ T-cell responses. One hypothesis that we explored in this work is that other cell subsets not efficiently targeted by our cre/lox model were still producing PGE$_2$. Certain cell subsets such as MZMs do not have effective gene deletion using the Lyz2$^+$ promoter to drive cre recombinase expression [33]. Our immunohistochemistry data suggest that MZMs still may be capable of producing PGE$_2$ in the COX-2$^{fl/fl}$ CD11c-cre$^+$ Lyz2-cre$^+$ mice (S6D Fig). For this reason, we hypothesized that subsets such as these may still be producing sufficient levels of PGE$_2$ to drive T-cell responses. One way to assess the role of MARCO$^+$ MZMs is use of a new cre recombinase-driving promoter, SIGN-R1, developed by Pirgova et al [34]. SIGN-R1 is a lectin binding receptor expressed on MZMs and drives more efficient deletion of genes by the cre/lox system [34]. Generation of triple COX-2 knockout mice that express the SIGN-R1-cre in combination with our reported COX-2$^{fl/fl}$ Lyz2-cre$^+$ CD11c-cre$^+$ model could be informative about the role of MZMs in production of PGE$_2$. Though we show that Lyz2$^+$ and CD11c$^+$ cells contribute to PGE$_2$, analysis of MZMs and other myeloid cells will further understanding of PGE$_2$ production.

The lack of diminished cell-mediated immunity could also be due to local acting effects of PGE$_2$. It is possible that even if PGE$_2$ levels are below detection at a whole spleen level, certain cells are able to produce PGE$_2$ locally in sufficient concentration to drive T-cell responses. More sensitive measures of PGE$_2$, such as quantitative mass spectrometry imaging recently developed, would be required to analyze local responses such as these [62]. These novel techniques enable analysis of location of PGE$_2$ and other eicosanoids within a spleen and could detect lower concentrations [62]. Similarly, the sensitivity of the receptor PGE$_2$ is acting upon during *L. monocytogenes* infection could influence how much PGE$_2$ is necessary for inducing a response. PGE$_2$ binds primarily to four receptors, EP1-4 [63]. EP3 and EP4 are higher affinity receptors (kD ~1nM compared to 10-15nM for EP1/2) [63,64]. Should the higher affinity receptors be identified as the important receptors for influencing immunity during *L. monocytogenes* infection, even lower concentrations of PGE$_2$ still induced in our COX-2$^{fl/fl}$ Lyz2-cre$^+$ CD11c-cre$^+$ model may be sufficient for cell-mediated responses. Further analysis as to relevant receptors and which cells they are expressed on could help elucidate these details.

Lastly, how PGE$_2$ facilitates T-cell responses in the context of *L. monocytogenes* immunization remains unknown. In innate immune cells, PGE$_2$ influences expression of co-stimulatory and activation markers. PGE$_2$ signaling in dendritic cells upregulates the co-stimulatory molecules OX40L and 4-1BBL [65], thereby promoting T-cell proliferation. Similarly, PGE$_2$ signaling in macrophages leads to polarization towards a more inflammatory M1 phenotype [66] and aids in activation [67]. Furthermore, PGE$_2$ promotes migration of innate cell subsets, leading to enhanced migration towards CCL21[68,69] and MCP-1 [70,71]. These proinflammatory functions suggest that PGE$_2$ may be acting to enhance immunity through its local effects on innate immune cells. PGE$_2$ may also be influencing immunity more directly on T-cell subsets, such as through polarization of T-cells towards a Th1 phenotype [72]. Additionally, PGE$_2$ leads to higher expression of OX-40L, OX-40, and CD70 directly on T-cells, promoting T-cell interactions and sustaining immune responses [65]. In order to more fully understand how PGE$_2$ facilitates T-cell responses to *L. monocytogenes*, a comprehensive analysis of these effects on both T-cells and innate immune cells is required.

We and others have shown that innate immune responses substantially influence cell-mediated immune responses, particularly the inflammatory milieu induced during infection. Here, we present evidence that one pathway critical for immunity, induction of PGE$_2$, is dependent on access to the cytosol. Furthermore, we show that PGE$_2$ is produced by macrophages and

dendritic cells. These data suggest analysis and modulation of eicosanoid levels, particularly PGE$_2$ levels, may be informative to improve the use of *L. monocytogenes*-based immunotherapeutic platforms.

## Materials and methods

### Ethics statement

This work was carried out in strict accordance with the recommendations in the Guide for the Care and Use of Laboratory Animals of the National Institutes of Health. All protocols were reviewed and approved by the University of Wisconsin-Madison Institutional Animal Care and Use Committee.

### Bacterial strains

The *Listeria monocytogenes* strains used in this study were all in the 10403s background. The attenuated (LADD) strain used in the analysis of T-cell responses was in the Δ*actA*Δ*inlB* background as previously described and engineered to express full length OVA and the B8R$_{20-27}$ epitope [30]. OVA and B8R$_{20-27}$ were constructed as a fusion protein under the control of the *actA* promoter with the secretion signal of the amino terminal 300bp of the ActA gene [15]. This fusion protein was integrated into the site-specific pPL2e vector as previously described [15].

### Mouse strains

Six- to eight-week-old C57BL/6 male and female mice were obtained from the NCI and Charles River NCI facility. *Ptgs2*$^{-/-}$ (COX-2$^{-/-}$) mice were obtained from Jackson Laboratory and maintained as heterozygote breeding pairs. *Ptges*$^{-/-}$ (mPGES1$^{-/-}$) mice lacking microsomal PGE synthase have been previously described [73–75]. In order to generate cell-type specific COX-2 knockout mice, COX-2$^{fl/fl}$ mice (stock number 030785) were obtained from Jackson Laboratory and crossed with Lyz2-cre (stock number 004781), CD11c-cre (stock number 008068), or CD4-cre expressing mice (stock number 022071), all also obtained from Jackson Laboratory. Double Lyz2-cre and CD11c-cre expressing mice were generated by crossing COX-2$^{fl/fl}$ Lyz2-cre$^+$ mice with COX-2$^{fl/fl}$ CD11c-cre$^+$ mice. Genotypes were confirmed by PCR using the primer pairs in Table 1. In experiments assessing PGE$_2$ production and T-cell responses in these mice, COX-2$^{flf/l}$ cre$^-$ mice were used as controls for initial experiments, followed by duplicate experiments using either COX-2$^{flf/l}$ cre$^-$ or WT C57BL/6 mice. Experiments utilizing COX-2$^{fl/fl}$ cre$^-$ mice or C57BL/6 mice induced consistent PGE$_2$ levels and T-cell responses, suggesting that insertion of the *loxP* site independently did not influence COX-2 expression.

**Table 1. Genotyping primers.**

| Mouse genotype | Forward (5'-3') | Reverse (5'-3') |
|---|---|---|
| COX-2$^{fl/fl}$ | AAT TAC TGC TGA AGC CCA CC | CTT CCC AGC TTT TGT AAC CAT |
| CD4-cre | GAACC TGATG GACAT GTTCA GG (cre specific) | AGTGC GTTCG AACGC TAGAG CCTGT (cre specific) |
| | TTACG TCCAT CGTGG ACAGC (internal control) | TGGGC TGGGT GTTAG CCTTA (internal control) |
| CD11c-cre | ACT TGG CAG CTG TCT CCA AG (cre specific) | GCG AAC ATC TTC AGG TTC TG (cre specific) |
| | CAA ATG TTG CTT GTC TGG TG (internal control) | GTC AGT CGA GTG CAC AGT TT (internal control) |
| Lyz2-cre | CCC AGA AAT GCC AGA TTA CG | CTT GGG CTG CCA GAA TTT CTC |

## BMDM and BMDC generation and infection

Bone marrow-derived macrophages and dendritic cells were made using six- to eight-week-old *Ptgs2*$^{-/-}$ (COX-2$^{-/-}$) or C57BL/6 mice as previously described [15,76]. Briefly, bone marrow was harvested and macrophages were cultured in the presence of M-CSF from transfected 3T3 cell supernatant for six days with a supplement of M-CSF at day three and frozen down for storage. Dendritic cells were cultured in the presence of 20ng/ml recombinant GM-CSF (BD Biosciences, San Jose, CA) for 7 days with a supplement of 20ng/mL GM-CSF every third day. For infection, BMDMs or BMDCs were plated at 1x10$^6$ cells/well in a 12 well dish overnight +/- 100ng/mL PAM3CSK4. The following morning, cells were infected with indicated strains of *L. monocytogenes* or PBS control at an MOI of 10 unless otherwise indicated. Thirty minutes later, supernatant was removed and replaced with medium containing 50μg/mL gentamycin to remove extracellular bacteria. At the indicated times, cells were harvested for western blot or qRT PCR and supernatant was harvested for eicosanoid analysis as described below.

## qRT PCR

RNA was isolated from BMDMs or BMDCs using the RNAqueous-Micro Total RNA Isolation Kit (Invitrogen), and DNAse treated with Turbo DNAse (Invitrogen) according to manufacturer's instructions. 500ng total RNA was reverse transcribed in 10μL reactions using the iScript cDNA Synthesis Kit (BioRad) according to manufacturer's instructions and cDNA was diluted 10-fold using molecular grade water (Invitrogen). 2.5μL diluted cDNA was used as template in a 10μL qRT-PCR reaction performed in duplicate using gene-specific primers and Kapa SYBR Green Universal qPCR mix (KAPA Biosystems) according to manufacturer's instructions using a BioRad CFX Connect Real-Time PCR System. The sequences of gene-specific primers are shown in Table 2. Data was analyzed using Excel and all RNA abundances were calculated by using a standard curve of synthesized template (Integrated DNA Technologies, G-Blocks) and are normalized to *ActB* (β-actin).

## Western blots

BMDMs or BMDCs were harvested and protein was extracted using the Pierce SDS-PAGE Sample Prep Kit (Thermo) according to the manufacturer's instructions. Total protein content was measured by the Pierce BCA Protein Assay Kit (Thermo) and equivalent protein levels were loaded into a polyacrylamide gel (BioRad). Samples were transferred onto a nitrocellulose membrane using a semi-dry transfer apparatus before blocking with a 5% skim milk solution for thirty minutes at room temperature. After washing 3x with PBS-T, the membrane was incubated overnight at 4°C with the primary antibodies anti-COX-2 (1:200, Cayman Chemical) and anti-β-actin loading control (1:1000, ThermoFisher) in a 5% bovine serum albumin solution. The following day samples were washed with PBS-T before being incubated with secondary antibodies (anti-rabbit 800 at 1:10,000, anti-mouse 680 at 1:5,000). Samples were

**Table 2. qRT PCR primers.**

| Gene | Forward (5'-3') | Reverse (5'-3') |
|------|-----------------|-----------------|
| *ActB* | TGGCATTGTTACCAACTGGGACG | GCTTCTCTTTGATGTCACGCACG |
| *Ptgs2* | TGAGCAACTATTCCAAACCAGC | GCACGTAGTCTTCGATCACTATC |
| *Ptges* | GGATGCGCTGAAACGTGGA | CAGGAATGAGTACACGAAGCC |
| *Pla2g4a* | CAGCACATTATAGTGGAACACCA | AGTGTCCAGCATATCGCCAAA |
| *Ifnb1* | GCACTGGGTGGAATGAGACTATTG | TTCTGAGGCATCAACTGACAGGTC |
| *Il1b* | GACCTGTTCTTTGAAGTTGACGG | TGTCGTTGCTTGGTTCTCCTTG |

imaged on a LiCor imager and analyzed via ImageStudio. Sample signal was normalized to β-actin and relative abundance was compared to wild-type *L. monocytogenes*.

### *In vivo* immunizations and pharmacological treatments

*L. monocytogenes* of the wild-type, attenuated (LADD), or *Δhly* background were grown overnight in brain heart infusion media at 30C. The bacteria were back diluted 1:5 and allowed to grow to log phase (OD0.4–0.6, ~1–1.5 hours) at 37C, with aeration, prior to infection. Bacteria were diluted in PBS and mice were infected with 200μL at the indicated doses intravenously. For bacterial burden analysis, mice were sacrificed at 12hpi and livers were homogenized in 0.1% Nonidet P-40 in PBS and plated on Luria-Bertani plates. For splenic macrophage depletion, 200μL clodronate, PBS control, or endosome lipid control (Encapsula Nano Sciences) were given intravenously 24 hours prior to bacterial infection according to the manufacturer's instructions. For B-cell depletion, mice were dosed intraperitoneally with 12.5μg of Ultra-LEAF Purified anti-mouse CD20 (clone SA271G2) or isotype control (clone RTK4530) in 100μL PBS 24 hours prior to bacterial infection. Depletion efficacy of relevant cell subsets was confirmed by assessing abundance of splenic CD11b$^+$ cells (clone M1/70), CD11c$^+$ cells (clone N418), or B220$^+$ cells (clone RA3-6B2) by flow cytometry. Celecoxib (Cayman Chemical) was milled into standard mouse chow (Envigo) at 100mg/kg and fed ad lib for 48 hours before and after immunization [16,77].

### Eicosanoid measurement

*In vivo* eicosanoid levels were assessed by mass spectrometry. *Ex vivo* analysis was assessed by mass spectrometry or ELISA, as indicated. For *ex vivo* PGE$_2$ analysis by ELISA, BMDMs were plated and infected as described above. Supernatant was harvested at indicated timepoints and diluted 1:100 in ELISA buffer (Cayman Chemical). PGE$_2$ was detected using a PGE$_2$ ELISA kit (Cayman Chemical) following the manufacturer's instructions. For mass spectrometry eicosanoid extractions, spleens from mice were harvested at twelve hours post immunization and flash frozen in tubes containing 50ng deuterated PGE$_2$ standard (Cayman Chemical) in 5μL methanol and stored overnight at -80C. For *ex vivo* extractions, 1mL of supernatant was flash frozen in tubes similarly containing 50ng deuterated PGE$_2$ standard in 5μL methanol. The following day, two mL of ice cold methanol were added to the tissue culture supernatant or spleens. Spleens were homogenized in glass homogenizers. Samples then were incubated at 4C for 30 minutes. Next, cellular debris was removed by centrifugation and samples were concentrated to 1mL volume before being acidified with pH 3.5 water and loaded onto conditioned solid phase C18 cartridges. Samples were washed with hexanes before eluting using methyl formate followed by methanol. Samples were concentrated using a steady stream of nitrogen gas and suspended into 55:45:0.1 MeOH:H$_2$O:acetic acid and analyzed on an HPLC coupled to a mass spectrometer (Q Exactive; Thermo Scientific) using a C18 Acquity BEH column (100mm x 2.1 mm x 1.7μm) operated in negative ionization mode. Samples were eluted with a mobile phase 55:45:0.1 MeOH:H$_2$O:acetic acid shifted to 98:2:0.1 over 20 minutes. Mass-to-charge ratios included were between 100 and 800 and compared to standards (Cayman Chemical) by analysis via MAVEN [78,79]. Samples were normalized to deuterated PGE$_2$ levels to ensure that differences between samples was not due to extraction efficiency. Average PGE$_2$ ion count for each sample was then compared to PBS or WT infected samples, as indicated.

### T-cell analysis

Mice were sacrificed seven days after immunization and splenocytes were isolated as previously described [6]. In brief, red blood cells were lysed using ACK buffer and then splenocytes were counted using a Z1 Coulter counter. For tetramer analysis, splenocytes were immediately

blocked for Fc (Tonbo Bioscience) and stained for B8R tetramer (AF488, 1:300, NIH Tetramer Facility, Atlanta, GA) followed by staining with anti-CD3 (PeCy7, 1:100, clone 145-2C11) and anti-CD8α (eFlour450, 1:200, clone 145-2C11). Cells were then stored overnight at 4C in a 1:1 of IC Fixation Buffer (ThermoFisher Scientific) and FACS buffer. For analysis of cytokine production, $1.7 \times 10^6$ cells were plated in a 96 well dish and incubated for five hours in the presence of $B8R_{20-27}$ (TSYKFESV) or $OVA_{257-264}$ (SIINFEKL) peptides and brefeldin A (eBioscience). Splenocytes were then subjected to FC block (Tonbo Bioscience) and stained with anti-CD3 (FITC, 1:200, clone 145-2C11) and anti-CD8α (eFlour450, 1:200, clone 53–6.7) before treatment with fixing and permeabilization buffers (eBioscience). Cells were then further stained with anti-IFNγ (APC, 1:300, clone XMG1.2). Samples were acquired using the LSRII flow cytometer (BD Biosciences, San Jose, CA) and analyzed with FlowJo software (Tree Star, Ashland, OR).

## Cryosection preparation and immunofluorescence microscopy of infected spleens

C57BL/6 or COX-2$^{fl/fl}$ CD11c-cre$^+$ Lyz2-cre$^+$ mice were infected intravenously by tail vein injection of $10^7$ LADD *L. monocytogenes* in 150 μl of PBS. Mice were sacrificed at 3 or 10 hpi and spleens harvested and snap frozen in OCT for immunofluorescence microscopy as described previously [22]. Uninfected mice were used as negative controls. Briefly, 5μm spleen cryosections were cut using a Leica CM1850 cryostat, mounted on Superfrost Plus microscope slides (Thermo Fisher) and stored at -80˚C until use. Slides were fixed in 10% buffered formalin phosphate at RT for 5 minutes and sections, washed in TBS and blocked with StartingBlock T20 Blocking buffer containing Fc blocker (Thermo Scientific, 37543). Sections were incubated with unconjugated *L. monocytogenes* monoclonal Ab (Invitrogen, MA1-20271), anti-MARCO polyclonal Goat IgG-Biotin (R&D Systems, BAF2956), and FITC-conjugated COX2 polyclonal antibody (Cayman Chemical, 10010096) at 1/100-200 dilution at RT for 1-2h in dark humidified incubation chamber or isotype control antibodies including Rabbit IgG-FITC (Invitrogen, 11-4614-80), Armenian Hamter IgG-PE (Invitrogen, 13-4888-81) and Mouse IgG2a kappa (Invitrogen, 14-4724-81). Biotinylated and unconjugated primary antibodies were detected by incubating with Streptavidin-PE (Pharmingen, 534061) and Rat anti-mouse IgG2a antibody (Invitrogen, 17-4210-80) respectively. Slides were preserved using ProLong Diamond Antifade mounting media (Invitrogen, P36965) and clear nail polish to seal the edges. Slides were analyzed using an Olympus IX51 fluorescence microscope equipped with LCPlanFL 20x/0.4 NA and UPlanFL 40x/1.3 oil objectives, an X-Cite 120 excitation unit (Exfo), FITC/PE/APC optimized filter sets (Semrock), an Orca Flash 2.8 monochrome camera (Hamamatsu) and SlideBook software (Intelligent Imaging Innovations) for hardware control and image acquisition. Images were captured with both 20x an 40x objective with exposure times ranging from 200-400ms. Pseudo-colored 3-channel RGB mages were imported into Imaris 9.6 (Bitplane) for smoothing, contrast enhancement (linear contrast stretch), annotation and colocalization analysis.

## Statistical analysis

Statistical analysis was performed by GraphPad Prism Software (La Jolla, CA) and analyzed via Mann Whitney *U* test or one-way ANOVA with Bonferroni's correction as indicated.

## Supporting information

**S1 Fig. Infection or priming of BMDMs and BMDCs does not alter cPLA2 transcript expression. Activation of type I IFN or inflammasomes does not substantially alter PGE₂**

**production.** BMDMs or BMDCs were infected with the indicated strains of *L. monocytogenes* at an MOI 10 +/- the TLR2 agonist PAM3CSK4 and assessed 6hpi for the expression of *Pla2g4a* (encoding cPLA2, A), *Ifnb1* (encoding interferon β, B), and *Il1b* (encoding IL-1β, C) by qRT PCR. PAM3CSK4-primed BMDMs were infected at an MOI 10 for six hours (unless otherwise specified) with the indicated strains of *L. monocytogenes*. PGE$_2$ levels in the supernatant was then assessed by ELISA (D-F). Data are a combination of three independent experiments. Significance was determined by a one-way ANOVA with Bonferroni's correction. $^*p < 0.05$, $^{**}p < 0.01$, $^{***}p < 0.001$, $^{****}p < 0.0001$.
(TIF)

**S2 Fig. Infection of BMDMs or BMDCs with *L. monocytogenes* does not lead to a global increase in eicosanoid production.** Wild-type or COX-2$^{-/-}$ BMDMs (A) or BMDCs (B) were infected with the indicated strains of *L. monocytogenes* at an MOI of 10 +/- the TLR2 agonist PAM3CSK4. Supernatant was harvested 6hpi and assessed for prostaglandin D$_2$ (PGD$_2$), thromboxane B$_2$ (TXB$_2$), or leukotriene B$_4$ (LTB$_4$). Data was normalized to d-PGE$_2$ and fold change is relative to PBS treated controls. Data are a combination of two independent experiments. Significance was determined by a one-way ANOVA with Bonferroni's correction. $^*p < 0.05$.
(TIF)

**S3 Fig. COX-2 deletion in CD4$^+$ cells does not alter PGE$_2$ production.** Indicated strains of mice were immunized with 10$^7$ LADD *L. monocytogenes* or PBS control. 12hpi spleens were harvested and assessed for PGE$_2$ by mass spectrometry. Data was normalized to d-PGE$_2$ levels and fold change is compared to PBS controls. Data are representative of two independent experiments. Significance was determined by a one-way ANOVA with Bonferroni's correction.
(TIF)

**S4 Fig. COX-2 deletion in CD11c$^+$ or Lyz2$^+$ does not impair T-cell responses.** Indicated strains of mice were infected with 10$^7$ LADD *L. monocytogenes*. 7dpi splenocytes were examined for OVA-specific CD8$^+$ T-cell responses. %IFNγ (A) or number IFNγ (B) per spleen was assessed. B8R-tetramer positive CD8$^+$ T-cell responses were also assessed (C). Data are a representative of two independent experiments of 4–5 mice per group. Significance was determined by a Mann-Whitney *U* test.
(TIF)

**S5 Fig. COX-2 deletion in both CD11c$^+$ and Lyz2$^+$ does not impair T-cell responses.** Indicated strains of mice were infected with 10$^7$ LADD *L. monocytogenes*. 7dpi splenocytes were examined for OVA-specific CD8$^+$ T-cell responses. %IFNγ (A) or number IFNγ (B) per spleen was assessed. B8R-tetramer positive CD8$^+$ T-cell responses were also assessed (C). Data shown are representative of two independent experiments of 3–5 mice per group. Significance was determined by a Mann-Whitney *U* test (B-C). $^*p < 0.05$.
(TIF)

**S6 Fig. Clodronate treatments reduces CD11b$^+$ and slightly reduces CD11c$^+$ populations in the spleen. Anti-CD20 treatment depletes splenic B220$^+$ cells, but does not influence PGE$_2$ production.** C57BL/6 mice were dosed with 200μL clodronate, liposome control (encapsome), or PBS 24 hours prior to immunization with 10$^7$ LADD *L. monocytogenes*. 12hpi spleens were harvested and assessed for CD11b$^+$ (A) and CD11c$^+$ (B) populations by flow cytometry. C57BL/6 mice were dosed with 50μg anti-CD20 or isotype control 24 hours prior to immunization with 10$^7$ LADD *L. monocytogenes*. 12hpi spleens were harvested and assessed

for B220$^+$ populations by flow cytometry (C) or PGE$_2$ levels by mass spectrometry (D). COX-2$^{fl/fl}$ CD11c-cre$^+$ Lyz2-cre$^+$ were infected with 10$^7$ LADD *L. monocytogenes*. 10hpi spleens were harvested, cryosections were cut, and sections were stained for *L. monocytogenes*, COX-2, and MARCO (E). Data shown are a combination of two independent experiments. Significance was determined by a Mann-Whitney *U* test. $^*p < 0.05$, $^{**}p < 0.01$.
(TIF)

## Acknowledgments

We would like to thank the NIH Tetramer Core Facility for provision of MHC-I B8R tetramers. Any opinions, findings, and conclusions or recommendations expressed in this material are those of the author(s) and do not necessarily reflect the views of the National Science Foundation.

## Author Contributions

**Conceptualization:** Courtney E. McDougal, Zachary T. Morrow, Seonyoung Kim, Drake Carter, Mark J. Miller, John-Demian Sauer.

**Formal analysis:** Courtney E. McDougal, Zachary T. Morrow, Tighe Christopher, Seonyoung Kim, Drake Carter, David M. Stevenson, Mark J. Miller, John-Demian Sauer.

**Funding acquisition:** Courtney E. McDougal, Zachary T. Morrow, Daniel Amador-Noguez, Mark J. Miller, John-Demian Sauer.

**Investigation:** Courtney E. McDougal, Zachary T. Morrow, Tighe Christopher, Seonyoung Kim, Drake Carter, Mark J. Miller, John-Demian Sauer.

**Methodology:** Courtney E. McDougal, Zachary T. Morrow, Tighe Christopher, Seonyoung Kim, Drake Carter, Daniel Amador-Noguez, Mark J. Miller, John-Demian Sauer.

**Project administration:** David M. Stevenson, Daniel Amador-Noguez, Mark J. Miller, John-Demian Sauer.

**Resources:** Daniel Amador-Noguez, Mark J. Miller, John-Demian Sauer.

**Supervision:** Daniel Amador-Noguez, Mark J. Miller, John-Demian Sauer.

**Validation:** Courtney E. McDougal, Zachary T. Morrow, Tighe Christopher, Seonyoung Kim, Drake Carter, Daniel Amador-Noguez, Mark J. Miller, John-Demian Sauer.

**Visualization:** Courtney E. McDougal, Seonyoung Kim, Drake Carter, Mark J. Miller, John-Demian Sauer.

**Writing – original draft:** Courtney E. McDougal, Zachary T. Morrow, Seonyoung Kim, Drake Carter, Mark J. Miller, John-Demian Sauer.

**Writing – review & editing:** Courtney E. McDougal, Zachary T. Morrow, Mark J. Miller, John-Demian Sauer.

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
