## [Decision Letter · Decision Letter 0]

23 Apr 2021

Dear Dr. Sauer,

Thank you very much for submitting your manuscript "Prostaglandin E2 induction by cytosolic Listeria monocytogenes in phagocytes is necessary for optimal T-cell priming" for consideration at PLOS Pathogens. As with all papers reviewed by the journal, your manuscript was reviewed by members of the editorial board and by several independent reviewers. In light of the reviews (below this email), we would like to invite the resubmission of a significantly-revised version that takes into account the reviewers' comments.

Consensus among all reviewers exists that the submitted work describes several intriguing findings and is potentially of substantial interest to the field. However, reviewers also agreed that the data shown here failed to conclusively demonstrate that phagocyte production of PGE2 triggered by cytosolic Listeria is important for induction of the CD8 responses and that the evidence in support of the proposed model is rather circumstantial. All reviewers suggested various experimental approaches to alleviate these concerns and to elevate the paper beyond what some deemed a somewhat preliminary study in its current form. Since all the proposed experiments in aggregate seem rather excessive, I will try to provide some guidance here as to what specific experimental work is likely required for a revised MS to be deemed appropriate for publication in PloS Pathogens (regardless, a point-by-point rebuttal should address all reviewers’ concerns):

- Please address comments 1a- 1e by reviewer 1- experimentally, as needed

- Both reviewers 2 & 3 felt that a delta-HLY strain expressing OVA or B8R was an important control missing from key experiments

- Address concerns that chlodronate-mediated depletion is a blunt tool and provide more direct evidence to demonstrate that phagocyte-derived PGE2 is critical for initiating the CD8+ T cell response in vivo (see suggestions by reviewer 2)

Since addressing these points will most likely require substantial experimental work, the deadline for the submission of a revised manuscript can be moved out as needed

We cannot make any decision about publication until we have seen the revised manuscript and your response to the reviewers' comments. Your revised manuscript is also likely to be sent to reviewers for further evaluation.

Sincerely,

Jorn Coers

Pearls Editor

PLOS Pathogens

Raphael Valdivia

Section Editor

PLOS Pathogens

Kasturi Haldar

Editor-in-Chief

PLOS Pathogens

orcid.org/0000-0001-5065-158X

Michael Malim

Editor-in-Chief

PLOS Pathogens

orcid.org/0000-0002-7699-2064

Consensus among all reviewers exists that the submitted work describes several intriguing findings and is potentially of substantial interest to the field. However, reviewers also agreed that the data shown here failed to conclusively demonstrate that phagocyte production of PGE2 triggered by cytosolic Listeria is important for induction of the CD8 responses and that the evidence in support of the proposed model is rather circumstantial. All reviewers suggested various experimental approaches to alleviate these concerns and to elevate the paper beyond what some deemed a somewhat preliminary study in its current form. Since all the proposed experiments in aggregate seem rather excessive, I will try to provide some guidance here as to what specific experimental work is likely required for a revised MS to be deemed appropriate for publication in PloS Pathogens (regardless, a point-by-point rebuttal should address all reviewers’ concerns):

- Please address comments 1a- 1e by reviewer 1- experimentally, as needed

- Both reviewers 2 & 3 felt that a delta-HLY strain expressing OVA or B8R was an important control missing from key experiments

- Address concerns that chlodronate-mediated depletion is a blunt tool and provide more direct evidence to demonstrate that phagocyte-derived PGE2 is critical for initiating the CD8+ T cell response in vivo (see suggestions by reviewer 2)

Since addressing these points will most likely require substantial experimental work, the deadline for the submission of a revised manuscript can be moved out as needed

Reviewer's Responses to Questions

**Part I - Summary**

Reviewer #1: The manuscript by McDougal et. al. addresses a fundamental question in microbial immunology to intracellular pathogens, i.e., what are the immune pathways that lead to cell-mediated immunity (CMI) characterized by the production of antigen specific CD8+ T-cells. The authors of this paper use the Listeria monocytogenes (Lm) model to explore the requirements for the induction of CMI. In this model, previous work from many labs have shown that sublethal doses of Lm induce a robust CD8+ T-cell response, while Lm mutants that lack listeriolysin O (LLO) fail to induce similar immunity. Although dogma has it that innate immune pathways lead to the induction of innate immunity, the specific pathways required have not been identified. In the case of Lm, it is known that wild-type bacteria activate cytosolic innate immune pathways including STING and the inflammasome to a limited extent, but neither pathway is necessary for induction of CMI. In a previous paper, these authors have provided evidence that prostaglandin E2 (PGE2) does contribute to the induction of CMI (ref 14). In the current submission, the authors ask if the production of PGE2 by macrophages and dendritic cells requires entry into the cytosol both in vitro and in vivo. They concluded that cytosolic entry is required for production of PGE2, although mice that are unable to make PGE2 in both macrophages and DCs still induce CD8 T-cells.

Reviewer #2: The authors report on efforts to follow up on their interesting previous finding that inhibition of COX2 or downstream production of PGE2 acts to suppress the induction of CD8+ T cell responses during Listeria monocytogenes infection (Ref 14 in the manuscript). The work here implicates Lyz2+ macrophages and Cd11c+ dendritic cells (DC) as sources of PGE2 in mice infected with L. monocytogenes. The authors further show PGE2 is only detected in spleens of the infected mice when L. monocytogenes expresses the LLO virulence factor. A bacterial strain lacking this factor (delta-HLY) was also reduced in its ability to induce PGE2 release from BMDMs and BMDCs infected in culture. Based on these findings, the authors propose that PGE2 production selectively occurs upon bacterial infection of the cytosol in these cells and thus the inability to induce PGE2 production may explain why delta-HLY L. monocytogenes fails to stimulate robust CD8+ T cell responses.

While the topic and findings are interesting the data provided fail to conclusively demonstrate that phagocyte production of PGE2 is important for induction of the CD8 responses. Indeed, the authors find that infection of mice in which COX2 (and therefore PGE2) are not produced by either Lyz2+ or CD11c+ cells has no observed effect on T cell activation. There is also no effort to experimentally address how cytosolic access of the bacterium regulated PGE2 production or the mechanism(s) by which PGE2 production acts to regulate CD8 T cell activation. Thus, in its present state, the paper is quite preliminary and incomplete.

Reviewer #3: McDougal et al. sought to understand the role of cytosolic Listeria monocytogenesis in the production of prostaglandin E2 and its effects in T cell priming. This is a well-written manuscript that utilizes various cell specific COX2 deletion, followed by infection with different strains of L. monocytogenesis. This is a potentially exciting manuscript, and the hypothesis that cytosolic scape is required for optimal PGE2 production in macrophages and DCs is indeed interesting. However, the authors did not deeply investigate the role of this bacterial strain in T cell priming. Also, it seems that there is a differential requirement of BMDCs and BMDMs in cytosolic Listeria-mediated PGE2 that was not discussed. Although the overall study is well-designed, many significant flaws decrease the enthusiasm for this work

**Part II – Major Issues: Key Experiments Required for Acceptance**

Reviewer #1: 1. My major criticism involves the in vitro analysis using cultured macrophages and DCs and their conclusion that PGE2 production is a result of Lm entering the cytosol. Host cells were infected with an equal amount of WT or LLO-minus Lm and PGE2 levels were evaluated after 6 hours. At this time point, there would be approximately 50-times more WT Lm than LLO-minus.

1a. I suggest that the authors show a time-course and identify the time of maximum PGE2.

1b. They should show the levels of PGE2 if more LLO-minus are added to be equivalent to the numbers of WT.

1c. Since PGE2 has a very short half-life, it is possible that PGE2 is induced early but is absent at the 6h time course.

1d. I am curious if they need to use mass spec for their analysis or would using commercially available ELISAs make this analysis easier?

1e. In the discussion, the authors mention STING, inflammasomes and type I IFN. It would be nice to see PGE2 production in mice lacking all of these.

1f. As the authors point out, PGE2 production involves calcium signaling and they suggest that LLO pore formation may be involved in triggering PGE2. They also mention a strain of Lm that it would be interesting if a strain engineered to delete LLO in the cytosol was evaluated (ref 51). I suggest they try this strain in vitro and possible in vivo as well.

2. the current study strongly suggest that PGE2 is induced by cytosolic access by Lm in phagocytes, leading to optimal CD8+ T-cell responses. However, although the authors demonstrate that cytosolic Lm induces PGE2 ex vivo and in vivo, the data showing that this process necessarily occurs in phagocytes for optimal T-cell priming is lacking. Part of challenge is, as the authors mention, that Clodronate treatment removes antigen presenting cells, making analysis of CD8+ T-cell priming irrelevant under those circumstances. As a result, the required experiments are not yet done, and would require mice lacking complete PGE2 induction in the MZMs. As an alternative, the authors look at MZMs as the possible source of PGE2 during wt Lm infection using IHC for colocalization studies (Fig5 c-d). While wt Lm does co-localize with MARCO and COX-2, the in vitro data demonstrate that both WT and LLO-minus Lm induce COX-2(fig 1C). Nevertheless, based on the current data and the previous literature, the results are highly suggestive that PGE2 is induced by cytosolic bacteria in phagocytes for optimal T-cell priming. However, this has not been completely demonstrated due to the removal of APCs by chlodronate, resulting in key CD8 + T-cell priming assays not being shown. The COX2 pathway in MZMs should be removed specifically, and T cells assayed after immunization.

3. To strengthen the paper, the authors should also bolster the concept that only the cytosolic strain induces PGE2 in MZMs, by comparing the LLO-minus and wt strains using the IHC methods already employed, to show that only cytosolic bacteria co-localize with MARCO and COX-2, whereas LLO-minus strains should not co-localize, providing further evidence that only cytosolic bacteria induce the PGE2 pathway in vivo. This work should not be too difficult or take too long to do since similar experiments were done already. The paper would significantly benefit from this figure since it is key to the main concept that LLO and wt induce PGE2 differentially in MZMs.

Reviewer #2: The authors outline in the Discussion a number of possible experiments that could move the studies forward and a variety of different directions. Given the paper's title and its focus on cytosolic induction of PGE2 synthesis by phagocytes, my opinion is that the authors should minimally seek to address the following:

(1) Provide additional data to more directly support that conclusion that phagocytes are an essential source of PGE2 production. Chlodronate depletion was shown to suppress measured PGE2 production, but in this is a blunt tool. It remains possible that such treatment suppresses PGE2 by non-phagocytes, e.g. by triggering massive induction of cell death? Can the authors demonstrate that depleting another cell population through a process that stimulates cell death (e.g. depletion of B cells with anti-CD20 or using a DTR/DT system) fails to eliminate measurable PGE2 in B6 or Cox-2FL/FL x Lyz2-cre + Cd11c-cre mice?

(2) Provide evidence that phagocyte PGE2 is critical for initiating the CD8+ T cell response. The lack of a reduction in CD8+ T cells in the Cox-2FL/FL x Lyz2-cre + Cd11c-cre mice could, as the authors propose, implicate that critical PGE2 can also be released by MZ macrophages. However, it is also possible that PGE2 from some other cell type is sufficient. Since chlodronate would be expected to also deplete DC, the best way to address this would be to crossing the floxed Cox2 allele onto a new cre driver that targets other phagocytes (and possibly crossing these mice again to the existing strain that combines Lyz2-cre and Cd11c-cre). The authors mention SIGNR1-cre in the Discussion. Mice that delete COX2 more broadly in macrophages could also be used (e.g. Csf1r-cre or Cd11b-cre). The authors might further consider using a cre driver that targets all hematopoietic cells to confirm that immune cells are key for PGE2 production that leads to the CD8+ T cell activation.

(3) Provide additional evidence to support the notion that PGE2 production requires access of L. monocytogenes to the macrophage cytosol. Although the delta-HLY strain did not trigger PGE2 in vivo, is seems that this strain was capable of triggering some PGE2 production in cultured BMDMs. This raises the possibility that the induction of PGE2 is not dependent on any specific cytosol detection mechanism and further that the lack of T cell priming by the delta-HLY strain could be due to some other defect. Indeed, HLY has been ascribed a number of functions in addition to promoting release of L. monocytogenes from vacuoles. It is also possible that elevated PGD2 or some other factor produced selectively during infection by the delta-HLY strain is responsible for the impaired T cell response. The authors should indirectly address this question by demonstrating if the triggering of PGE2 production by WT L. monocytogenes in vitro (and in vivo?) is dependent (or not) on inflammasome activation by the WT (but not delta-HLY) bacteria. It would also be useful to include experiments mentioned in the discussion using bacterial strains which have reduced ability to trigger inflammasome activation to see if these also have reduced ability to stimulate PGE2 and T cell responses. Even if these results show that the inflammasome is not required, they will provide useful information and advance the impact of the paper. Finally, important controls missing here include confirmation that a delta-HLY strain expressing OVA or B8R fails to activate the respective T cells and, assuming not, to show if the presence of the mutant strain does or does not suppress PGE2 production and T cell priming in mice that are also infected with WT bacteria.

Reviewer #3: The responses of BMDM and BMDCs to �hly is substantially different. In Fig 1B and 1C, BMDMs seem to be more responsive to �hly than BMDC. Yet, the expression of COX2 is not decreased in PAM-challenged BMDM, but it is in BMDCs. Yet, in Fig 1D, the �hly elicits decreased PGE2 production in both cell lines. What is the basis for such discrepancy?

Fig.2 should be combined with Fig.1 or 3. It does not provide enough information to be a stand-alone fig. Also, a better rationale for Fig 2B should be provided.

In Fig.3, why the authors used the LADD strain instead of �hly??? Using the LADD strain instead of �hly is not a fair comparison with the experiments in Figs 1 and 2 and does not provide evidence about the role of cytosolic Listeria in PGE2 production. To help a non-T cell expert understand the data, a rationale for the investigation of CD8 producing IFN, rather than CD4, should be provided.

Fig 5 is indeed very interesting. However, panels with higher magnification and resolution should be provided.

**Part III – Minor Issues: Editorial and Data Presentation Modifications**

Reviewer #1: I agree that there is now circumstantial evidence that phagocytes produce PGE2 and that PGE2 is necessary for full induction of CMI implying that the two are connected. However, I suggest that the authors change the title so as not to imply that they showed that phagocytes producing PGE2 were necessary for T-cell priming.

Reviewer #2: 1. Several references are not formatted properly. Lack page numbers, etc.

2. Figure 2 shows that bacterial burdens of the delta-HLY strain are not lower in the liver, but PGE2 is measured in the spleens. The authors should include cfu data from the spleens as well.

3. Given the partial reduction in PGE2 and lack of impaired T response, it seems important to confirm efficiency of COX2 deletion in CX2FL/FL x CD11c-cre and/or Lyz2-cre mice. For example, by showing there is no COX2 protein and PGE2 production by phagocytes isolated from these mice. Perhaps this information will also be relevant in addressing some of the concerns above.

4. At least some of the control mice for the PGE2 experiments should be COX2FL/FL mice. Perhaps the flox itself impairs COX 2 expression and thus PGE2 irrespective of the CD11c or Lyz2 cre?

5. I recommend the authors not use "Cd11c-cre" or "Lyz2-cre" as abbreviations for the cre x flox strains as this is confusing for the reader.

Reviewer #3: Also, the absolute concentration of PGE2 should be shown instead of fold changes. This reviewer understands that transforming the values for fold change might be required due to intrinsic variations in concentration, but the raw numbers of the basal levels and challenge with WT bacterial should be mentioned in the fig. legends. Is the n=2 of the experiments? If so, it is not appropriate to perform statistical analysis in n=2

In the introduction (page 5, lines 92, 93), please change prostaglandin synthases to prostanoid synthases. Prostaglandin synthase only applies to PGE2 production.

PLOS authors have the option to publish the peer review history of their article (what does this mean?). If published, this will include your full peer review and any attached files.

Reviewer #1: No

Reviewer #2: No

Reviewer #3: No
---

## [Decision Letter · Decision Letter 1]

8 Sep 2021

Dear Dr. Sauer,

We are pleased to inform you that your manuscript 'Phagocytes produce prostaglandin E2 in response to cytosolic Listeria monocytogenes' has been provisionally accepted for publication in PLOS Pathogens.

Best regards,

Jorn Coers

Pearls Editor

PLOS Pathogens

Raphael Valdivia

Section Editor

PLOS Pathogens

Kasturi Haldar

Editor-in-Chief

PLOS Pathogens

orcid.org/0000-0001-5065-158X

Michael Malim

Editor-in-Chief

PLOS Pathogens

orcid.org/0000-0002-7699-2064

Reviewer Comments (if any, and for reference):

Reviewer's Responses to Questions

**Part I - Summary**

Reviewer #1: The authors have satisfactorily addressed all of my comments.

Reviewer #2: In this revised manuscript, the authors have included new data to address previous concerns and toned down their conclusions to present a focused study that provides interesting new information to suggest phagocytic cells are the major source of PGE2 production during Listeria infection and reveal insights into the regulation of the PGE2 production.

Reviewer #3: I have no further comments

**Part II – Major Issues: Key Experiments Required for Acceptance**

Reviewer #1: (No Response)

Reviewer #2: no further concerns

Reviewer #3: I have no further comments

**Part III – Minor Issues: Editorial and Data Presentation Modifications**

Reviewer #1: (No Response)

Reviewer #2: line 312 - recommend substituting "ex vivo" with "in vitro"

line 340 - more accurate to say that PGE2 production "was still observed"

Reviewer #3: (No Response)

PLOS authors have the option to publish the peer review history of their article (what does this mean?). If published, this will include your full peer review and any attached files.

Reviewer #1: No

Reviewer #2: No

Reviewer #3: No

---

## [Editor Report · Acceptance letter]

17 Sep 2021

Dear Dr. Sauer,

We are delighted to inform you that your manuscript, "Phagocytes produce prostaglandin E2 in response to cytosolic Listeria monocytogenes," has been formally accepted for publication in PLOS Pathogens.

Best regards,

Kasturi Haldar

Editor-in-Chief

PLOS Pathogens

orcid.org/0000-0001-5065-158X

Michael Malim

Editor-in-Chief

PLOS Pathogens

orcid.org/0000-0002-7699-2064